# The role of glacier changes and threshold definition in the characterisation of future streamflow droughts in glacierised catchments

Marit Van Tiel[1,2,a], Adriaan J. Teuling[2], Niko Wanders[3,4], Marc J.P. Vis[5], Kerstin Stahl[6], and Anne F. Van Loon[1]

[1]School of Geography, Earth and Environmental Sciences, University of Birmingham, Birmingham, UK
[2]Hydrology and Quantitative Water Management Group, Wageningen University, Wageningen, The Netherlands
[3]Department of Civil and Environmental Engineering, Princeton University, Princeton, USA
[4]Department of Physical Geography, Utrecht University, Utrecht, The Netherlands
[5]Department of Geography, University of Zurich, Zurich, Switzerland
[6]Faculty of Environment and Natural Resources, University of Freiburg, Freiburg, Germany
[a]now at: Faculty of Environment and Natural Resources, University of Freiburg, Freiburg, Germany

*Correspondence to:* Marit Van Tiel (marit.van.tiel@hydrology.uni-freiburg.de)

**Abstract.** Glaciers are essential hydrological reservoirs, storing and releasing water at various time scales. Short-term variability in glacier melt is one of the causes of streamflow droughts, here defined as deficiencies from the flow regime. Streamflow droughts in glacierised catchments have a wide range of interlinked causing factors related to precipitation and temperature on short and long time scales. Climate change affects glacier storage capacity, with resulting consequences for discharge regimes and streamflow drought. Future projections of streamflow drought in glacierised basins can, however, strongly depend on the modelling strategies and analysis approaches applied. Here, we examine the effect of different approaches, concerning the glacier modelling and the drought threshold, on the characterisation of streamflow droughts in glacierised catchments. Streamflow is simulated with the HBV-light model for two case study catchments, the Nigardsbreen catchment in Norway and the Wolverine catchment in Alaska, and two future climate change scenarios (RCP 4.5 and RCP 8.5). Two types of glacier modelling are applied, a constant and dynamic glacier area conceptualisation. Streamflow droughts are identified with the variable threshold level method and their characteristics are compared between two periods, a historical (1975-2004) and future (2071-2100) period. Two existing threshold approaches to define future droughts are employed, 1) the threshold from the historical period and 2) a transient threshold approach, whereby the threshold adapts every year in the future to the changing regimes. Results show that drought characteristics differ among the combinations of glacier area modelling and thresholds. The historical threshold combined with a dynamic glacier area projects extreme increases in drought severity in the future, caused by the regime shift due to a reduction in glacier area. The historical threshold combined with a constant glacier area results in a drastic decrease of the number of droughts. The drought characteristics between future and historic periods are more similar when the transient threshold is used, for both glacier area conceptualisations. With the transient threshold causing factors of future droughts, can be analysed. This study revealed the different effects of methodological choices on future streamflow drought projections and it highlights how the options can be used to analyse different aspects of future droughts: the transient threshold

for analysing future drought processes, the historical threshold to assess changes between periods, the constant glacier area to analyse the effect of short term climate variability on droughts and the dynamic glacier area to model more realistic future discharges under climate change.

## 1  Introduction

Glaciers and snow packs are an important freshwater resource, supplying water to more than one-sixth of the Earth's population (Barnett et al., 2005). Glaciers play an essential role in the global water cycle as hydrologic reservoirs on various time scales (Jansson et al., 2003; Vaughan et al., 2013). They for example reduce the inter-annual variability by storing water in cold and wet years and releasing it in warm and dry years (Jansson et al., 2003; Koboltschnig et al., 2007; Zappa and Kan, 2007; Viviroli et al., 2011). Also on seasonal time scales glacier storage and release are important: the glacier melt peak in summer sustains discharge during otherwise low flow conditions (due to low precipitation or high evapotranspiration; e.g. Fountain and Tangborn, 1985; Miller et al., 2012; Bliss et al., 2014) and especially during low flow conditions downstream (Huss, 2011). Fluctuations in the summer glacier melt peak may therefore be an important driver of streamflow drought.

Drought is defined as a below-normal water availability (Tallaksen and Van Lanen, 2004; Sheffield and Wood, 2012) and streamflow drought (also called hydrological drought) is a drought in river discharge. According to this definition we defined streamflow droughts in this study as anomalies (or deficiencies) from the hydrological regime, including the important high flow melt season. Streamflow droughts are a recurring and worldwide phenomenon (Tallaksen and Van Lanen, 2004) which can have severe impacts on river ecology, water supply and energy production (e.g. Jonsdottir et al., 2005; van Vliet et al., 2016). Hydrological drought is often caused by meteorological drought (deficit in precipitation) which propagates through the hydrological cycle (Tallaksen and Van Lanen, 2004; Van Loon, 2015). In cold climates, where snow and ice are an important part of the seasonal water balance, streamflow drought can also be caused by anomalies in temperature (Van Loon et al., 2015). In glacierised catchments, 'glacier melt droughts', defined as a deficiency in the glacier melt peak and caused by below normal temperatures in the summer season (Van Loon et al., 2015), can be important to downstream water users.

Climate change is expected to have large influences on both glaciers and streamflow droughts due to a reduction in the water storage capacity of glaciers and snow packs. This will have major consequences for the water supply downstream (e.g. Kaser et al., 2010; Immerzeel et al., 2010; Huss, 2011; Finger et al., 2012). The Intergovernmental Panel on Climate Change (IPCC) reports with high confidence that glaciers worldwide are shrinking and that current glacier extents are out of balance with the current climate, indicating that glaciers will continue to shrink (Vaughan et al., 2013). Retreating glaciers affect the discharge regimes in glacierised catchments. Déry et al. (2009) and Bard et al. (2015) found a shift in the melt peak towards an earlier moment in the season in trend studies of observed streamflow, in British Columbia, Canada and in the European Alps, respectively. Also for the future, changes in the timing of the melt peak are expected, together with a more dominant role of rainfall and less snow accumulation (Horton et al., 2006; Jeelani et al., 2012, for Swiss Alps and Western Himalyas, respectively). Two recent studies showed that retreating glaciers can have contrasting effects on the hydrology. Ragettli et al. (2016) project rising flows with limited shifts in the seasonality for the Langtang catchment in Nepal and a reduced and shifted

peak in streamflow for the Juncal catchment in Chile. The latter was also found by Lutz et al. (2016) for the Upper Indus basin. Farinotti et al. (2012) show the combined responses with increasing and then decreasing annual discharges for several glacierized catchments in Switzerland by modelling the period 1900-2100. What these projected changes in glacial hydrology mean for streamflow droughts has, however, not been explicitly modelled. From global and continental scale drought studies, we expect streamflow droughts to become more severe in the future (Bates et al., 2008; Van Huijgevoort et al., 2014), with an increase in average streamflow drought duration and deficit volume expected for the globe (Van Huijgevoort et al., 2014; Wanders and Van Lanen, 2015). Also for Europe, Feyen and Dankers (2009) and Forzieri et al. (2014) found that many river basins are likely to experience more severe streamflow drought.

These projections are however strongly dependent on the methodology applied in the analysis and for both future glacier modelling and future drought analysis many options exist. In order to make projections for hydrology in glacierised catchments under climate change, a glacio-hydrological model is needed. Especially in highly glacierised catchments and when modelling long time periods, a realistic representation of the glacier in the model is crucial. However, complex ice flow models require a lot of input data (e.g. glacier bathymetry and density estimates, see also Immerzeel et al., 2012; Naz et al., 2014) which is often not available and they are in general not applicable for hydrological modelling (Huss, 2011). Different types of glacier geometry conceptualisations are therefore used in hydrological studies. For example, past studies by Klok et al. (2001), Verbunt et al. (2003), and Schaefli et al. (2005) used a simple infinite and constant glacier reservoir in their hydrological model. Also e.g. Akhtar et al. (2008), Tecklenburg et al. (2012) and Sun et al. (2015) used a constant glacier area in their modelling studies, as a benchmark to compare with model runs where the glacier area is adjusted. Juen et al. (2007) simulate future glacier extent assuming a new steady state in the future obtained by reducing the glacier area gradually until the future annual mass balance is zero. Stahl et al. (2008) used a volume-area relation to re-scale the glacier based on modelled glacier mass balances, however distributing the area reduction only conceptually in space. Huss et al. (2010) used a more detailed glacier representation in their model by introducing the △h-parametrisation to calculate the transient evolution of the glacier surface elevation and area. Huss et al. (2010) found that the simulation of glacier evolution with this △h-parametrisation method was comparable to the results of a 3-D finite-element ice flow model. Li et al. (2015) used the approach of Huss et al. (2010) in combination with the well known HBV model (Bergström et al., 1995; Seibert and Vis, 2012). The effect of these different glacier area conceptualisations on streamflow drought characterisation remains to be investigated.

For the analysis of future streamflow drought methodological questions have been raised in the literature that relate to the definition of drought as a below-normal water availability. To quantify below-normal discharge often a threshold method is used that defines the 'normal' based on a baseline period. In the large scale drought studies mentioned above (Feyen and Dankers, 2009; Wanders and Van Lanen, 2015; Van Huijgevoort et al., 2014; Forzieri et al., 2014) and e.g. also in Wong et al. (2011); Lehner et al. (2006); Arnell (1999), a threshold based on a historical period was used to define streamflow droughts in the future. It can be questioned if this historical threshold is a good indicator of the 'normal water availability' in the future (see Wanders et al., 2015; Wanders and Wada, 2015; Van Loon et al., 2016). Especially in cold climates, expected regime shifts lead to the identification of severe droughts if evaluated against a historical threshold (Van Huijgevoort et al., 2014). This is particularly relevant in studies on future changes in streamflow drought in glacierised catchments where we expect fast changing regimes

due to the retreat of glaciers (e.g. Horton et al., 2006; Lutz et al., 2016). Wanders et al. (2015) therefore developed a transient threshold approach that takes into account changing regimes under climate change. This transient threshold assumes adaptation to long-term changes in the hydrological regime and hence identifies future streamflow droughts with reference to changed normal conditions. Wanders et al. (2015) applied this method to identify future streamflow droughts on a global scale and found that it reduces the area for which an increase in drought duration and deficit volume are expected from 62% to 27%. The transient threshold approach has however never been tested at the catchment scale and more specifically not in glacierised catchments.

This study aims to systematically test the effect of different methodological choices in simulating and analysing streamflow drought in glacierised catchments and elucidate which method to use for which purposes. We focus on two options for glacier modelling in a hydrological model (constant and dynamic glacier area) and two different drought threshold approaches (historical and transient threshold) resulting in four combinations. We test these combinations in two contrasting case study catchments in Norway and Alaska and discuss the implications for projections of future streamflow drought in glacierised basins in general.

## 2 Study areas and data

### 2.1 Study areas

Two catchments, one in Alaska (the Wolverine catchment) and one in Norway (the Nigardsbreen catchment), are used as case study in this research (Fig. 1), because of their good data availability, especially regarding glaciological data. The catchments are highly glacierised; i.e. 67% (for the Wolverine catchment) and 70% (for the Nigardsbreen catchment). The Wolverine glacier is a so called "benchmark glacier", where a long-term glacier monitoring program is maintained by the United States Geological Survey (USGS, 2015). Annual mass balances of the Wolverine glacier are negative since 1990. The glacier has a southerly aspect. The area of the Wolverine catchment is 25 $km^2$ and the catchment elevation range is 360–1700 m. It is located in the Kenai mountains in Alaska and close to the ocean at 60°N. It experiences a maritime climate (O?Neel et al., 2014). Long term average monthly temperatures range from $-6.7$°C to $+8.8$°C. The catchment receives most of its annual precipitation (2700 mm) in autumn (410 mm in September) and precipitation is lowest in summer (100 mm in June). The Nigardsbreen glacier in Norway is one of the largest outlet glaciers of the Jostedalsbreen, which is the largest glacier in Europe. The Nigardsbreen glacier shows alternating negative and positive annual mass balances, however the cumulative mass balance series is positive and shows an increasing trend since around 1990. The main aspect of the glacier is south-east. The catchment area is 65 $km^2$ and it has a large elevation range of 260–1950 m. The climate of this catchment is also maritime. Long term average monthly temperatures range from $-6.6$°C to $+6.6$°C. Precipitation amounts are highest in winter (450 mm in December) and lowest in spring (130 mm in May). Annual precipitation is around 3300 mm. The discharge station is located at the outlet of lake Nigardsbrevatn.

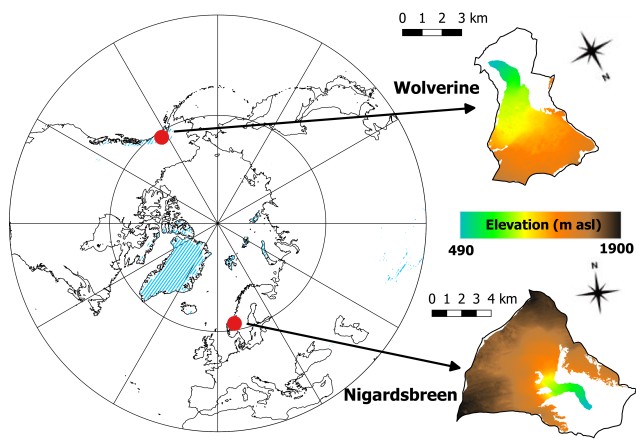

**Figure 1.** Location of case study catchments in Alaska (Wolverine catchment) and Norway (Nigardsbreen catchment). The coloured part in the catchments indicate the glacier areas of 2006 (Wolverine) and 2009 (Nigardsbreen) and the elevation of the glaciers. The light blue colour in the overview map shows glaciated areas.

## 2.2 Climate and hydrometric data

Observations of temperature ($T_{obs}$) and precipitation ($P_{obs}$) were used to force the model in the calibration and validation period and to validate the climate model data in the historical period. Daily $T_{obs}$ and $P_{obs}$ data of Nigardsbreen catchment were taken from a gridded dataset based on interpolation of observations from different gauging stations. From this dataset the

catchment average precipitation and temperature were calculated by the Norwegian Water Resources and Energy Directorate (NVE). Data were available for the period 1957-2014. Daily $T_{obs}$ and $P_{obs}$ data of the Wolverine catchment were obtained from USGS and were available for the period 1967-2015. The data comes from a weather station close to the margin of the Wolverine glacier. However, the Wolverine catchment is a windy site, where windspeeds up to 100 km/h can occur during precipitation events, which can result in an under catch problem. Therefore, after comparison with ERA-Interim precipitation

data (Dee et al., 2011), observed precipitation amounts were increased with a factor 2.5, to account for this precipitation under catch in the Wolverine catchment. This was verified during the calibration process where the model forced with increased precipitation amounts resulted in a better fit with observed discharge than using the original precipitation values. Gaps in the $T_{obs}$ time series (7%) of the Wolverine catchment were filled in with linear interpolation (for < 10 days missing data) or, for longer than 10 days missing data, with data from surrounding National Oceanic and Atmospheric Administration (NOAA)

stations (taking into account altitude differences) or, when no data was available from surrounding stations, with long term average daily temperatures. Gaps in the $P_{obs}$ time series (7%) of Wolverine were filled based on surrounding NOAA stations, again accounting for elevation differences.

    For the future projections, daily P and T data from a set of climate models were used ($P_{cm}$ and $T_{cm}$). Additionally the model in the historical period was forced with climate model data, in order to compare discharge and droughts between the

historical and future period. The climate model data is output from GCM-RCM model combinations from the World Climate

Research Program Coordinated Regional Downscaling Experiment (CORDEX Giorgi et al., 2009). For Norway data from EURO-CORDEX and for Alaska data from the North-America CORDEX (Jacob et al., 2014) were available. The resolution of the data over Norway is $0.11°$ and for Alaska $0.22°$. Nearest neighbour interpolation to the center point of the catchments was used to obtain catchment average $P_{cm}$ and $T_{cm}$ from the climate models. Climate model data for the period 1975-2004 (historical period) was used as reference data and was compared with $P_{obs}$ and $T_{obs}$. For the period 2006-2100, the climate model outcomes for two climate scenarios were used, i.e. the RCP 4.5 and RCP 8.5 scenarios. For Norway, bias corrected (with E-OBS, Haylock et al., 2008) climate model output data from eight GCM-RCM model combinations were available for the RCP 4.5 scenario and nine for the historical period and the RCP 8.5 scenario. For Alaska only data from one GCM-RCM model combination was available, without bias-correction. Therefore, the empirical quantile mapping method was applied to perform bias correction on the Alaskan data by using the observations of the weather station in the Wolverine catchment (Teutschbein and Seibert, 2012). This method was chosen because it is the same method as was used for the Norwegian climate data.

Observed discharge ($Q_{obs}$) was used for calibration and validation of the model and was provided by NVE and USGS, for Nigardsbreen and Wolverine (USGS Waterdata, 2016), respectively. The discharge was measured at the outlet of the catchments. Daily discharge data was available for 1963–2013 for Nigardsbreen and 1969–2015 for the Wolverine catchment. In the Wolverine discharge time series, gaps were present of several years. These years were excluded from the analysis.

## 2.3   Glaciological data

Seasonal glacier-wide mass balances of both glaciers were also obtained from USGS (O?Neel et al., 2016) and NVE (Andreassen and Engeset, 2016). The mass balances were used for calibration of the HBV-light model. Geodetically adjusted seasonal mass balances (winter and summer mass balances) were available for the Wolverine glacier and a homogenised seasonal mass balance series was available to this study for the Nigardsbreen glacier (Van Beusekom et al., 2010; O?Neel, 2014; Andreassen and Engeset, 2016).

Glacier outlines were used to define the glacier fraction in the catchments. These glacier outlines were obtained from the Randolph Glacier Inventory (RGI Version 5.0 Pfeffer et al., 2014) and from NVE (Winsvold et al., 2014; Andreassen et al., 2012). The glacier outlines were also used in combination with ice thickness data to define the volume of the glaciers. The ice thickness maps were available at a spatial resolution of $100 \times 100$ m for Nigardsbreen and $25 \times 25$ m for Wolverine. The information on distributed ice thickness of the glaciers from the maps was used for the dynamic glacier area modelling. For the Wolverine glacier the ice thickness data of Huss and Farinotti (2012), and for the Nigardsbreen glacier the data of Andreassen et al. (2015) were used.

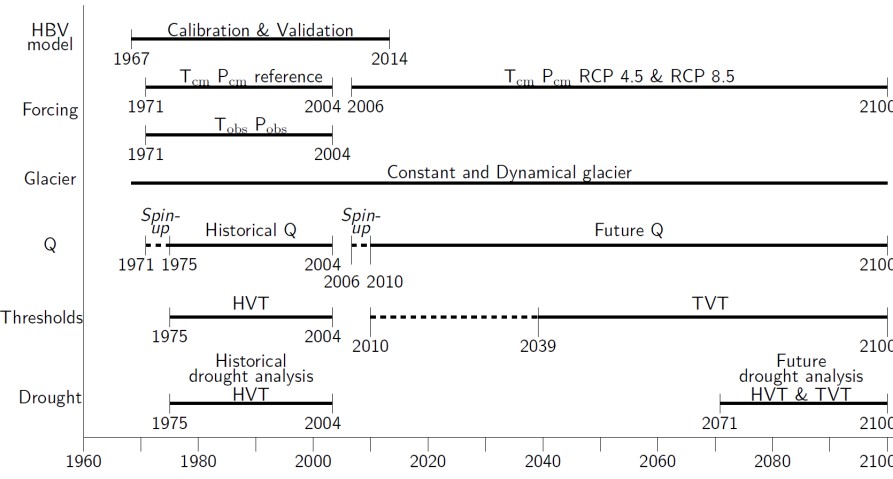

**Figure 2.** Timeline indicating the simulation periods, forcings and periods for threshold derivation and application. The different glacier area conceptualisations, constant and dynamic, are used in all simulation periods.

## 3  Methods

### 3.1  General modelling approach

The main variable of interest in this research is the river discharge. Since we are interested in the future, streamflow is modelled using a coupled glacio-hydrological model (see Sect. 3.2). Streamflow droughts are studied in two periods, a historical period (1975-2004) and a future period (2071-2100), in order to assess changes in drought characteristics between both periods (see Fig. 2). To systematically test the effect of glacier dynamics and threshold approach on future streamflow droughts and their characteristics, four scenarios, in which the glacier dynamics and threshold approach options are combined, are used to characterise streamflow droughts in these two periods (Fig. 3). The Historical Variable Threshold (HVT) and the Constant glacier area conceptualisation (C) represent the baseline conditions which we compare with the changing conditions: the Transient Variable threshold (TVT) and Dynamical glacier area conceptualisation (D) (see Fig. 3).

   The two threshold approaches that are tested and compared in our glacierised case study catchments are the more often used historical variable threshold (HVT) method, based on a fixed reference period in the past, and the recent introduced transient variable threshold (TVT) method, based on a changing reference period, thereby taking into account changes in the hydrological regime. The calculations of the thresholds are explained in Sect. 3.5. The glacier modelling options that are evaluated include a static and infinite glacier reservoir and a glacier geometry change conceptualisation using the $\triangle$h-parametrisation of Huss et al. (2010). These two glacier modelling options, in the following referred to as 'constant' and 'dynamic' glacier modelling options, are further explained in Sect. 3.2. Although the constant glacier modelling option will be unrealistic in transient mode we include this option in our analysis because dynamical glacier modelling is not yet included in all (large) scale hydrological models (e.g. Zhang et al., 2013) and it is an interesting benchmark, also frequently used in other studies (Akhtar et al., 2008;

**Figure 3.** Four analysis scenarios. The matrix shows the combination of the two threshold approaches with the two different glacier area conceptualisations, resulting in four possible combinations. The baseline options are indicated in black and the options where changes are taken into account are shown in red.

Stahl et al., 2008; Tecklenburg et al., 2012). The effect on streamflow drought characterisation has not yet been assessed. $P_{obs}$, $T_{obs}$ and $Q_{obs}$ were used to calibrate and validate the model, and were compared with simulated discharge ($Q_{sim}$) obtained by forcing with observations ($Q_{sim_o}$) and climate model data ($Q_{sim_{cm}}$) in the historical period, to address the uncertainty in both components. Future runs start in 2006 with a 4 year spin-up period, so that discharge is modelled for the period 2010-2100, to

include the transient evolution of the glacier area during the 21st century (Fig. 2). The model is forced with RCP 4.5 and RCP 8.5 climate change scenario data during the future simulations.

## 3.2   Conceptual model

The model used in this study is the conceptual HBV-light model with extended glacier routine (Seibert and Vis, 2012; Seibert et al., 2017). It is a version of the original HBV model developed at the Swedish Meteorological and Hydrological institute

(Bergström et al., 1995). The model is semi-distributed, based on elevation zones, vegetation zones and aspect classes. Daily temperature and precipitation and daily or long term monthly potential evapotranspiration are needed as input variables. The model simulates discharge and also calculates the contributions of the different components (rain, glacier ice (Q_g) and snow) to the total discharge. A glacier profile, in which the ice volume in the different elevation zones is defined, is needed in order to run the model with a dynamic glacier area.

The model consists of different routines. The glacier, snow and soil moisture routine are semi-distributed, whereas the groundwater and routing routine are lumped. The model simulates discharge at a daily time step. Based on a threshold temperature, precipitation will fall either as snow or rain. A snowfall correction factor is used in the model to multiply the snowfall

with to compensate for systematic errors in snowfall measurements and for evaporation/sublimation from the snowpack (not explicitly modelled). In the snow and glacier routine the melt is computed by a degree-day-method. A different degree-day factor is used for snow and glacier, because of the lower albedo of glacier ice. Snow redistribution is not taken into account. For a detailed model description we refer to Seibert and Vis (2012). The calibrated parameter values of the snow and glacier

routine are presented in Appendix A. The glacier in the model is represented by two components: a glacier ice reservoir and a glacier water content reservoir. A small fraction (0.001) of the snow on the glacier is transformed into ice each time step. When the glacier is not covered by snow, glacier melt is taking place for temperatures above the threshold temperature. Glacier melt is added to the glacier water content reservoir, just like water from snow on the glacier which melts and rain falling on the glacier. From the glacier water content reservoir, water is flowing directly into the routing routine. The amount of discharge

from the glacier is based on an outflow coefficient which varies in time because it depends on the snow water equivalent of the snowpack on the glacier. It represents the development of glacial drainage systems (Stahl et al., 2008). In the non-glaciated part of the catchment snow melt and rainfall flow into the soil routine. From here water can evaporate or be added to the groundwater reservoirs. Peak flow, intermediate flow and baseflow discharge components are generated within the groundwater routine, which is followed by the routing routine, in which the total discharge of one timestep is distributed over one or

multiple timesteps according to a weighting function.

The glacier routine in the HBV-light model can be used as a static or dynamic conceptualisation of the glacier in the catchment. In the static conceptualisation the glacier area is constant over time, while in the dynamic conceptualisation the area of the glacier is adjusted every year. The dynamic glacier conceptualisation in the HBV-light model is based on Huss et al. (2010), who proposed a simple parametrisation to calculate the change in glacier surface elevation and area ($\triangle$h-parametrisation), so

that future glacier geometry change can be approximated without using complex ice flow modelling. The $\triangle$h-parametrisation describes the spatial distribution of the glacier surface elevation change in response to a change in mass balance and has also been used in other studies e.g. Salzmann et al. (2012); Farinotti et al. (2012); Li et al. (2015); Duethmann et al. (2015). The implementation of various dynamic glacier change options into HBV-light based on the $\triangle$h-parametrisation are described and tested in Seibert et al. (2017). In HBV-light one out of three possible type curves for different glacier sizes can be chosen

(Huss et al., 2010). Furthermore, a glacier profile, in which the water equivalent and area of the glacier for each elevation band (elevation bands are subdivisions of the elevation zones) are specified, is required by HBV-light as input for the dynamic glacier conceptualisation. Before the actual model simulation starts, the glacier profile is melted in steps of 1% of the total glacier volume, and for each step the $\triangle$h-parametrisation of Huss et al. (2010) is applied to compute the areal change for each elevation zone. This information is stored by the HBV-light model in a lookup table of percentage of melt and corresponding

glacier areas. This table is then used to dynamically change the glacier during the actual model simulation. Each hydrological year the area of the glacier is updated by calculating the percentage of glacier volume change from the modelled mass balance and selecting the corresponding glacier areas from the lookup table.

## 3.3 Model set up

For daily temperature and precipitation input, we used observations or output from climate models. The HBV-light model requires a climate station at a certain elevation for the input of P and T. For Wolverine the HBV climate station elevation was set to the elevation of the weather station in the catchment for $T_{obs}$, $P_{obs}$, $T_{cm}$ and $P_{cm}$. For the Nigardsbreen catchment the average catchment elevation was used for the HBV climate station elevation for $T_{obs}$ and $P_{obs}$ and the average elevation of the RCM model grids for the $T_{cm}$ $P_{cm}$. P and T values for each elevation zone are calculated based on precipitation and temperature lapse rates, which are calibration parameters. Monthly evapotranspiration (E) was calculated for all simulation periods with the Blaney-Criddle method by using monthly average temperatures in order to get E values for both the historical and future simulations (Xu and Singh, 2001; Brouwer and Heibloem, 1986). The monthly values were linearly interpolated to retrieve daily values which were used as input to HBV-light. Each catchment was divided in several elevation zones, with elevation bins of 100 or 200 m depending on the elevation range in the catchment. Each elevation zone was split up in three aspect classes (North, South, East-West). The mean elevation of each elevation zone and the area of each elevation zone and aspect class were determined from the ASTER Digital Elevation Model (DEM). Missing values present in the ASTER DEM of Nigardsbreen were filled in by interpolation. The lake present in the Nigardsbreen catchment was defined as separate model unit.

To determine the glacier area in each elevation zone, glacier outlines of 2006 were used. For the static glacier conceptualisation these areas were used in all model runs, independent of time. However, in order to run the model with the dynamical glacier settings, initial glacier areas and glacier profiles were adapted to the largest glacier extent within the specific simulation period. For the future simulation period, it was assumed that the glacier extent will be largest at the start of the period (2006). Therefore initial glacier areas and the glacier profile based on ice thickness maps and 2006 glacier outlines needed no adaptation. For the other simulation periods (historical period, calibration period and validation period), the largest glacier extent was determined from area information from USGS for the Wolverine glacier and from homogenised area data from NVE for Nigardsbreen glacier (Andreassen and Engeset, 2016). The 2006 glacier areas and glacier profiles were adapted to these largest glacier extents.

For the construction of the 2006 glacier profile each glacier elevation zone was subdivided in smaller elevation bands, with elevation bins of 20 or 50 m, depending on the size of the elevation zone. For each elevation band the average ice thickness was determined from the ice thickness maps and converted into millimeter water equivalent (mm w.e.q.) by multiplying with the ratio of the densities from ice to water (0.917). The adjustment of the glacier profile to another glacier extent was done based on volume-area scaling (Bahr et al., 1997; Andreassen et al., 2015) to calculate the needed increase in ice thickness/water equivalent to match the new volume based on the new largest glacier extent. When the largest glacier extent did not occur at the beginning of the simulation period, an initial glacier fraction was defined in the glacier profile which was also calculated with the volume-area scaling method.

### 3.4 Calibration

The models were calibrated against (selected periods of) observed discharge and seasonal mass balances using the automatic calibration tool Genetic Algorithm and Powell (GAP) optimization in HBV-light (Seibert and Vis, 2012; Seibert, 2000). Including mass balances in the calibration is known to improve the model performance significantly (Konz and Seibert, 2010; Mayr et al., 2013; Engelhardt et al., 2014). For each catchment the model was calibrated with a constant glacier area conceptualisation and a dynamic glacier conceptualisation, so that a different parameter set was obtained for both glacier area conceptualisations. To calibrate on mass balances, the dates of maximum and minimum mass balances were used for the winter balance and the summer balance, respectively for the Wolverine catchment, and the actual measurement dates of the summer and winter balances for the Nigardsbreen catchment (meta data from NVE). A calibration period of at least 10 years was used for both catchments. The objective function that was maximised during the calibration is

$$R = 0.4 \times R_{\text{effG}} + 0.4 \times R_{\text{effS}} + 0.2 \times R_{\text{effP}} \tag{1}$$

with

$$R_{\text{eff}} = 1 - \left( \frac{\sum (Obs - Sim)^2}{\sum (Obs - \overline{Obs})^2} \right),$$

where $R$ is the model performance, $R_{\text{effG}}$ the calibration on glacier mass balances, $R_{\text{effS}}$ the calibration on the discharge from April-September and $R_{\text{effP}}$ is the calibration on the peak discharges. $Obs$ and $Sim$ are observed and simulated (seasonal) discharge or glacier mass balances, respectively. A $R_{\text{eff}}$ value of one indicates a perfect fit for that variable.

After the calibration was performed, model performance was evaluated with the Kling-Gupta Efficiency (KGE) which is defined as:

$$KGE = 1 - \sqrt{(r-1)^2 + (\alpha - 1)^2 + (\beta - 1)^2} \tag{2}$$

In Eq. 2 $r$ is the Pearson product-moment correlation coefficient, $\alpha$ the ratio of the standard deviations of simulated and observed discharge and $\beta$ the ratio between the means of simulated and observed discharge (Gupta et al., 2009). A KGE value of one indicates a perfect fit between modelled and observed discharge.

### 3.5 Drought thresholds

A variable threshold level method was used to identify droughts and to determine their characteristics (Hisdal et al., 2000; Fleig et al., 2006; Van Loon, 2013). A drought occurs when a variable (in our study discharge) falls below the threshold. We used a daily variable threshold that is derived from a 30-day moving average discharge time series. The moving average time series was used to compute the daily flow duration curves and to determine the 80th percentile for use as a drought threshold (Van Loon et al., 2014). Usually threshold levels between the 70th and 95th percentiles are applied in drought studies (Fleig

et al., 2006). Using another threshold or different moving window size will result in slightly different drought characteristics but the percentile choice has less effect on the results when only looking at changes in drought characteristics and comparing different approaches, as was done in this study.

This variable threshold was calculated for both catchments and glacier conceptualisations separately. The historical variable threshold was calculated from the discharge in the historical period (1975-2004). For the future period two threshold approaches were used: 1) the variable threshold from the historical period (HVT) following the work of Wanders and Van Lanen (2015) and Van Huijgevoort et al. (2014), and 2) a transient variable threshold (TVT) that assumes adaptation in the future based on reduced or increased water availability of the preceding 30 year period (Wanders et al., 2015). Hence, each year in the future has a different TVT, calculated from the previous 30 years of discharge as described above. The same HVT was used in the historical period and the future period for both climate change scenarios. The TVT was used in the future period, but for both climate change scenarios (RCP 4.5 and RCP 8.5) a different transient threshold was calculated. For the Nigardsbreen catchment the multi-model mean $Q_{sim_{cm}}$ was used for calculation of the thresholds and the drought analysis.

We computed the drought duration, deficit and intensity, to characterise changes in drought characteristics. The drought duration is defined as the consecutive number of days that the discharge is below the threshold. Droughts with a duration of three days or shorter were not taken into account (Fleig et al., 2006). The drought deficit volume is computed by taking the cumulative difference between the drought threshold and the discharge, for each drought event. Drought intensity is defined as the deficit divided by the duration. We analysed drought processes by studying temperature, precipitation, snow water equivalent (SWE) and the different discharge components together with the total discharge following the approach of Van Loon and Van Lanen (2012) and Van Loon et al. (2015). The thresholds for these variables were computed in the same way as was done for the discharge, except for temperature for which we used the median as threshold.

## 4    Results

### 4.1    Calibration and validation of model and data

The KGE of the calibration and validation periods are generally high (Table 1). Especially the Nigardsbreen catchment shows a very good agreement between modelled and observed discharge (KGE=0.94). The KGE is slightly lower in the validation period of the Nigardsbreen catchment and somewhat higher for the Wolverine catchment. The latter might be caused by the very short validation period of Wolverine. The type of glacier area modelling does not influence the model performance with respect to discharge in both the calibration and validation period. The individual $R_{eff}$'s of Eq. 1 range between 0.51 and 0.90 for the seasonal calibration and between 0.15 and 0.60 for the peak flow calibration for the two catchments. The $R_{eff}$'s for the mass balance calibration are 0.51 and 0.83 for the dynamic glacier simulations of Nigardsbreen and Wolverine, respectively. The hydrological regimes of observed and modelled discharge also match well for Nigardsbreen for the historical period (Fig. 4a), for both types of forcing: observations and climate model data. For Wolverine only three years of observed data were available in the historical period, resulting in a more uncertain observed regime compared to the simulated regimes in Fig. 4d. The inset in Fig. 4d shows the matching observed regime and the simulated regime forced by observations for the calibration

**Table 1.** Model performance for the two catchments. Performance is expressed by KGE between observed and modelled discharge and shown for the calibration and validation periods and the dynamic (D) and constant (C) glacier area conceptualisations.

| | Calibration | | | Validation | | |
|---|---|---|---|---|---|---|
| Catchment | C | D | period | C | D | period |
| **Nigardsbreen** | 0.94 | 0.94 | 1967-2003 | 0.90 | 0.90 | 2004-2013 |
| **Wolverine** | 0.82 | 0.83 | 2005-2014 | 0.89 | 0.87 | 1973-1977 |

**Table 2.** Streamflow drought characteristics of observed and simulated discharge. Drought characteristics are shown for observed discharge (obs) and simulated discharge (sim) with constant (C) and dynamic (D) glacier area conceptualisation in the calibration period of Nigardsbreen and Wolverine.

| Catchment | Droughts in | Number | Avg. Duration [d] | Avg. Deficit [mm] | Avg. Intensity [mm/d] |
|---|---|---|---|---|---|
| Nigardsbreen | obs | 357 | 12.21 | 16.48 | 1.39 |
| (1967-2003) | sim-C | 565 | 9.66 | 12.27 | 1.23 |
| | sim-D | 484 | 10.92 | 13.40 | 1.27 |
| Wolverine | obs | 99 | 13.49 | 25.97 | 2.80 |
| (2005-2014) | sim-C | 114 | 13.89 | 19.28 | 2.02 |
| | sim-D | 99 | 12.95 | 25.73 | 2.64 |

period. Besides matching regimes, the model is also able to simulate a similar inter-annual variability in discharge compared to the observations for Nigardsbreen (Fig. 4b, historical period) and Wolverine (Fig. 4e, calibration period).

We also compared modelled and observed glacier mass balances for the dynamic glacier area (see Fig. 4c and 4f). During the calibration period the negative trend in cumulative mass balance is simulated very well by the model for the Wolverine catch-
5 ment (Fig. 4f). Winter mass balances and the total volume change are slightly underestimated. In the Nigardsbreen catchment the model simulates negative cumulative mass balances at the start of the calibration period, while observed cumulative mass balances are positive. In this period, the model did not capture the sign of the almost balanced conditions right. However, during the second half of the calibration period, the positive trend in mass balance is the same in the observations and simulations. The intra-annual differences in summer and winter balances are smaller in the simulations, in both catchments.
10 Finally, we verified the streamflow drought characteristics of observed and simulated discharge in the calibration period (Table 2). The number of droughts for Nigardsbreen are a bit higher in the simulations than in the observations. However, in general drought characteristics of observed and simulated discharge agree well for both catchments.

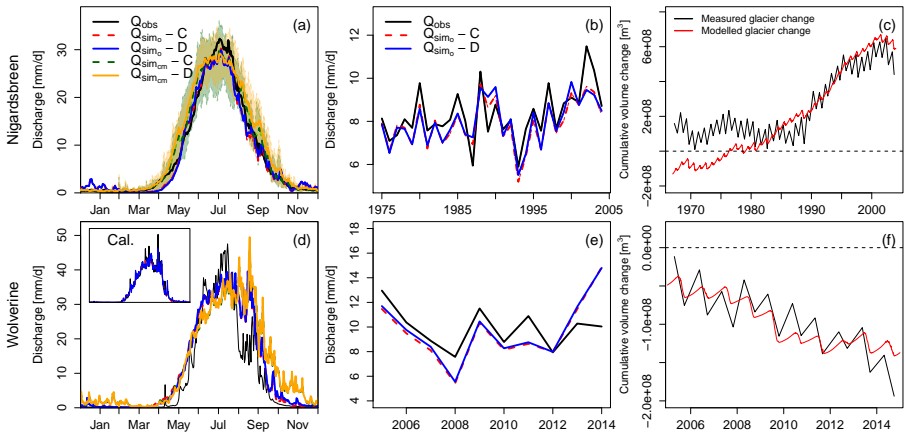

**Figure 4.** Model validation. The hydrological regime (a and d), annual discharges (b and e) and mass balances (c and f) are shown for Nigardsbreen catchment (a,b and c) and for Wolverine catchment (d, e and f). Panels a and d show results for the historical period (1975-2004) in order to compare observed discharge with both ($Q_{sim_o}$) and ($Q_{sim_{cm}}$) for both glacier model conceptualisations. The coloured areas in panel a indicate the range of discharge outputs as a result of the different climate models forcing. The inset in d shows the agreement between $Q_{obs}$ and $Q_{sim_o}$ for the Wolverine catchment during the calibration period. In panel b $Q_{sim_{cm}}$ is not shown because climate models only statistically represent historic climate. The inter-annual variability is shown for the historical period for Nigardsbreen and for the calibration period for Wolverine (due to $Q_{obs}$ availability). Panels c and f show the observed and measured glacier volume changes (water equivalent) for the calibration period of the Nigardsbreen and Wolverine glaciers, respectively.

## 4.2 Glacier area conceptualisations and their effect on discharge

During the constant glacier area runs, the model used a glacier area from 2006, both in the historical and future period (Fig. 5). Assuming that glaciers will shrink in the future, this area is too big during the future period and too small during the historical period because both glaciers had a larger area in the past compared to 2006. With a dynamic glacier area conceptualisation, this mismatch should not occur. In the Wolverine catchment, the glacier area in the historical period for the dynamic settings is indeed higher than the glacier area in the constant settings and the glacier area at the end of the historical period agrees with the constant area (observed glacier area in 2006) used throughout the whole modelled time period (Fig. 5). However, in the Nigardsbreen catchment the average modelled glacier area at the end of the historical period (2004) is smaller than the observed glacier area in 2006 (the constant glacier area). The model simulates a glacier area that decreases too much or a too small glacier extent was used at the start of the historical period and therefore there is a small jump between the average glacier area at the end of the historical period and the start of the future period (2006-2100) (the model periods are not coupled) (Fig. 5). The model simulates a glacier disappearance in the Wolverine catchment in the future when dynamic glacier areas are used, first in the RCP 8.5 scenario and later also in the RCP 4.5 scenario. In the Nigardsbreen catchment the glacier area develops similarly in both climate scenarios until 2060, after which the glacier is projected to shrink more quickly in the RCP

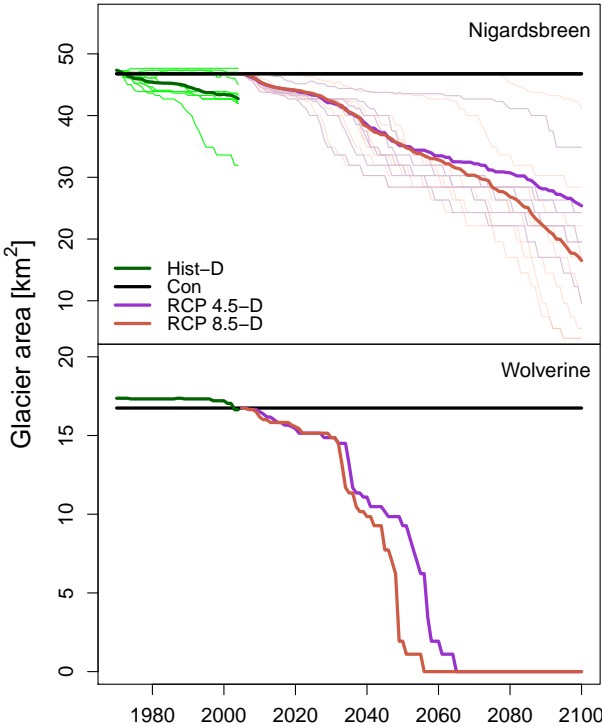

**Figure 5.** Temporal evolution of glacier areas for the historical and future period. The top panel shows the glacier areas for Nigardsbreen and the bottom panel for Wolverine. The glacier area in both glacier conceptualisations is shown. The lighter coloured lines in the Nigardsbreen graph for the historical period and the two RCP scenarios show the results of glacier area evolution for the different climate models forcing.

8.5 scenario. The spread in glacier area evolution projections for the Nigardsbreen catchment is however large. One climate model forcing even gives hardly any decrease in glacier area.

The different options for glacier area modelling have an effect on the future water availability (Fig. 6). The constant glacier area causes an amplification of the hydrological regime and increasing annual discharges in the future in both catchments. On the other hand, the dynamic glacier area causes a drastic change in the regime in the Wolverine catchment in the future period (2071-2100) (Fig. 6c). The regime in the Nigardsbreen catchment changes as well: the magnitude of the high flow period is smaller, the rising limb starts earlier and the recession limb starts later and is less steep than during the historical period. For both catchments the changes compared to the historical period are larger for the RCP 8.5 scenario. Annual discharges are projected to decrease in the Wolverine catchment with dynamic glacier area. The changes in multi-model mean annual discharges for Nigardsbreen are not so clear and the spread among the discharges forced by the different climate models increases in the future.

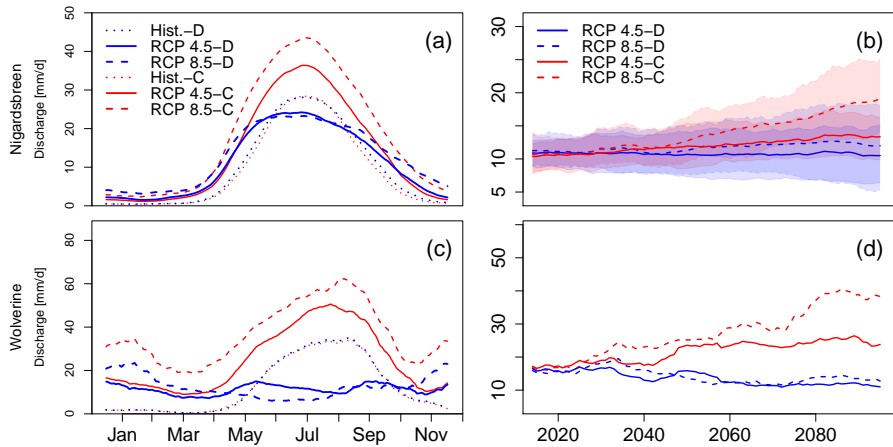

**Figure 6.** Future water availability. The left panels show the hydrological regime (30 day moving window of the daily average of 2071-2100) for Nigardsbreen (a) and Wolverine (c) and the right panels the annual average discharges for the future period (2010-2100) with a 10 year moving window (b: Nigardsbreen and d: Wolverine). Discharge is shown for both glacier area conceptualisations (colours) and both climate change scenarios (line type). The shaded areas in b indicate the spread in annual average discharges among the different climate models forcing.

### 4.3 Drought thresholds: the result of different glacier conceptualisations and threshold methods

Four approaches were used for the determination of the drought thresholds and future drought analysis, based on combinations of the threshold options and the glacier area conceptualisations. For both catchments the HVT-C and HVT-D thresholds are quite comparable (Fig. 7), except in the rising and recession limb of Nigardsbreen, where the HVT-D is above the HVT-C.

The transient thresholds however, vary in time. The magnitude of the high flow season in the TVT-C increases, while with the TVT-D it decreases each year in the future. In the Nigardsbreen catchment, the TVT-D threshold has a higher peak during the first decade compared to the historical period, after which the peak in the threshold becomes lower. All future TVT-D have however a longer high flow season than the historical threshold has. In the Wolverine catchment the TVT-D only shows a higher peak in August and September in the first years in the future compared to the HVT-D. Moreover, a shift is visible for

the rising limb in the TVT-D towards an earlier moment in the spring season for Nigardsbreen. The TVT-C develops in both catchments differently, in Nigardsbreen the peak shifts to earlier in the season, while for Wolverine the TVT-C peak shifts to later in the season.

    The transient threshold does not adapt at a constant rate, shown by the different spaces between the lines (Fig 7). The threshold follows the climate. The RCP 8.5 scenario gives similar results (not shown), but there is even more difference between

consecutive thresholds. This is due to a faster changing climate and discharge. For the Wolverine catchment the changes in the transient threshold are more extreme than Nigardsbreen, especially in the first half of this future period (2039-2070) of the TVT-D, in which the glacier is rapidly shrinking. Furthermore, due to the drastically changing regime, the transient threshold in

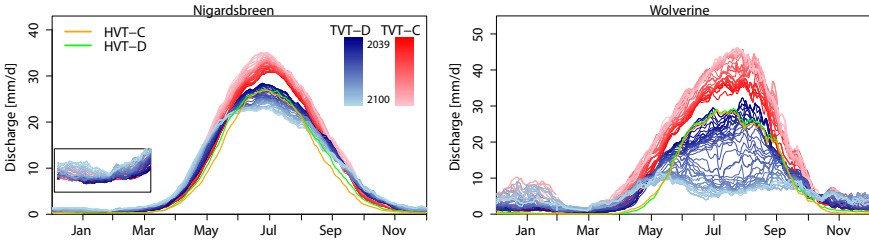

**Figure 7.** Drought thresholds for the four different scenarios HVT-C, HVT-D, TVT-C and TVT-D. The colour gradient for both transient thresholds (blue and red) indicates the adaptation of the threshold each year (for 2039-2100). The thresholds are shown for the Nigardsbreen (left) and Wolverine (right) catchments for climate scenario RCP 4.5. The inset in the left panel zooms in to the low flow period of Nigardsbreen.

the Wolverine catchment changes also rapidly in the historical low flow periods (winter), in contrast with Nigardsbreen where the threshold stays low in the historical low flow periods.

## 4.4 Effect of thresholds on the identification and characterisation of future droughts

Applying the different thresholds to the discharge time series shows when droughts (below threshold discharges) occur during the year (Fig. 8 shows an example for Nigardsbreen catchment). The HVT-C and TVT-C are applied to the discharge output of the model simulated with a constant glacier area conceptualisation and the HVT-D and TVT-D to the output produced with a dynamic glacier area conceptualisation. Applying the threshold of the past to the discharge of the future with a constant glacier area (HVT-C) results in (almost) no droughts (Fig. 8), due to increased glacier melt. If the threshold of the past is applied to discharge with a dynamic glacier area conceptualisation (HVT-D), severe droughts occur at the period of the threshold high flow season and in the recession limb of the discharge curves, due to a lower peak flow and a shift in the hydrological regime (Fig. 8).

Using the transient threshold results in future droughts with much smaller deficits volume, compared to droughts determined with HVT-D (Fig. 8). Droughts do not only occur in the peak flow period, but are more distributed over the season and occur in the rising limb and low flow period as well, in both the TVT-C and TVT-D case. In Fig. 8 streamflow droughts look more severe (higher deficits) in the TVT-D settings than TVT-C settings, while in both cases the threshold has adapted. This is probably caused by the contribution of glacier melt to discharge. In the TVT-D, the threshold is based on 30 previous years when the glacier was larger than the year to which the threshold is applied, resulting in droughts partly caused by glacier retreat. The TVT-C, on the other hand, is based on 30 previous years in which the climate was colder than the year the threshold is applied, resulting in less melt from the glacier compared to the year the threshold is applied (glacier area is constant) and consequently less droughts are observed in the high flow season compared to TVT-D.

Besides a different timing of streamflow droughts in the year, the four threshold scenarios also resulted in different drought characteristics (e.g. deficit volume). Comparing drought characteristics between historical and future periods shows the changes

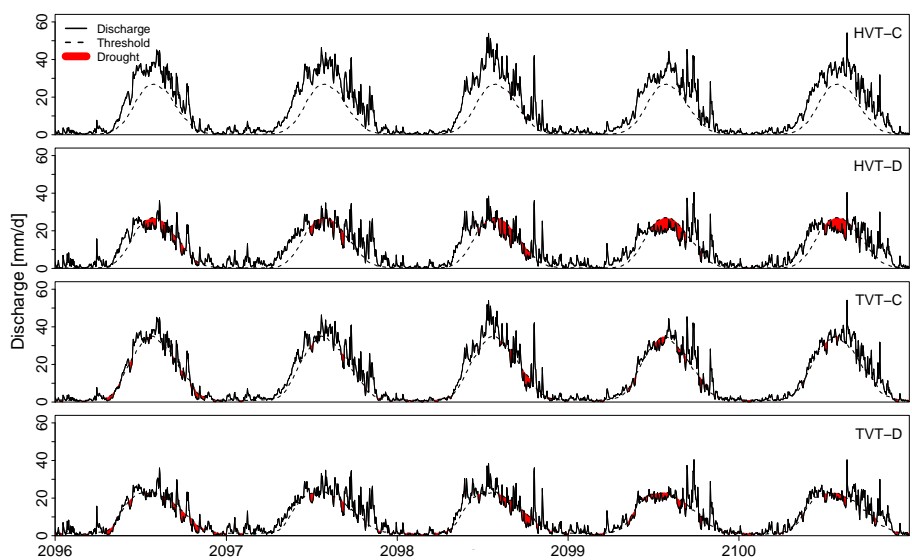

**Figure 8.** Example timeseries of possible timing and deficit volume of droughts in the four scenarios HVT-C, HVT-D, TVT-C and TVT-D. The droughts are shown for the Nigardsbreen catchment and the RCP 4.5 climate scenario for the period 2096-2100.

that can be expected in the future. However, the four scenarios resulted in different future changes in drought characteristics (Table 3). The number of droughts will decrease in both catchments when the HVT is used. The number of droughts will increase in the Wolverine catchment when the transient threshold is used. In Nigardsbreen catchment the TVT-C indicates a decrease in the number of droughts and the TVT-D only results in a small increase in the number of droughts. The average duration will only increase in the HVT-D scenario (except RCP 4.5 for Nigardsbreen), in the other threshold scenarios the average duration is projected to decrease. The HVT-D and TVT-D result in a projected increase in deficit volumes, except for TVT-D in the Wolverine catchment. However, deficit volumes are projected to increase more drastically when HVT-D is used. The HVT-C causes in general a decrease in deficit volume, while the TVT-C causes an increase in the deficit volume. Average intensities are in general projected to increase for all scenarios, with one exception for both Nigardsbreen and Wolverine (see Table 3). For most threshold scenarios the RCP 8.5 will give a larger change in the drought characteristic than the RCP 4.5 scenario compared to the historical period.

## 4.5 Effect of thresholds on analysing future drought processes

Using the four different methodological scenarios we can analyse streamflow drought processes differently. We separated the four scenarios in two comparisons: the glacier dynamics effect and the influence of the threshold approach on analysing drought processes. To study the glacier dynamics effect, the transient threshold was used for both glacier area conceptualisations (Fig. 9). No historical variable threshold was used here to exclude the effect of changing peak flow discharges compared to the historical period. The thresholds in the left and right panels of Fig. 9, are therefore based on the 30 previous years of discharge

**Table 3.** Change in drought characteristics in the future compared to the historical period. The percentages show the increase or decrease of the respective drought characteristic with respect to the historical period for each catchment and each glacier area conceptualisation.

| | Period | Scenario | Number | | Avg. Duration [d] | | Avg. Deficit [mm] | | Avg. Intensity [mm/d] | |
|---|---|---|---|---|---|---|---|---|---|---|
| | | | RCP 4.5 | RCP 8.5 | RCP 4.5 | RCP 8.5 | RCP 4.5 | RCP 8.5 | RCP 4.5 | RCP 8.5 |
| | **Hist** | **HVT + C** | 477 | | 7.86 | | 8.77 | | 0.96 | |
| | Fut | HVT + C | −80 % | −97 % | −39 % | −46 % | −61 % | −81 % | −35 % | −60 % |
| Nigardsbreen | Fut | TVT + C | −10 % | −47 % | −14 % | −26 % | 25 % | −3 % | 58 % | 48 % |
| | **Hist** | **HVT + D** | 467 | | 8.06 | | 9.34 | | 1.04 | |
| | Fut | HVT + D | −37 % | −58 % | −4 % | 12 % | 166 % | 309 % | 137 % | 191 % |
| | Fut | TVT + D | 9 % | 3 % | −15 % | −18 % | 38 % | 66 % | 63 % | 97 % |
| | **Hist** | **HVT + C** | 400 | | 10.21 | | 26.68 | | 3.21 | |
| | Fut | HVT + C | −66 % | −81 % | −35 % | −39 % | 9 % | −14 % | 53 % | 23 % |
| Wolverine | Fut | TVT + C | 23 % | 21 % | −21 % | −38 % | 79 % | 88 % | 106 % | 142 % |
| | **Hist** | **HVT + D** | 354 | | 10.1 | | 34.1 | | 4.39 | |
| | Fut | HVT + D | −21 % | −31 % | 25 % | 66 % | 431 % | 674 % | 133 % | 152 % |
| | Fut | TVT + D | 72 % | 91 % | −12 % | −12 % | −20 % | −13 % | −30 % | −21 % |

(TVT). In the constant glacier area conceptualisation a drought occurs in streamflow in the beginning of September, while for the dynamic glacier area several streamflow droughts occur between June and September (Fig. 9). The long term climatic changes cause the glacier to retreat in the future in the dynamic glacier conceptualisation. This glacier retreat can have an indirect effect on the occurrence of streamflow droughts because of less melt due to a smaller glacier. Streamflow droughts occurring in the summer period of 2092 in the Nigardsbreen catchment for the dynamic glacier area show this process (Fig. 9 right panel). Streamflow droughts are caused by short term (seasonal) anomalies in P (deficits) and T (lower) and additionally due to a retreating glacier resulting in less discharge from the glacier (Fig. 9). In the constant glacier area conceptualisation the effect of long term climate changes on glacier size is neglected and streamflow droughts are caused by short term climate variability. In Fig. 9 (left panel) the drought in September is caused by below normal temperatures, resulting in a deficit in $Q\_g$ and a drought in the total streamflow (Q). Furthermore, Fig. 9 (left panels) shows that glacier melt in summer is buffering against the propagation of precipitation deficits. This effect gets lost with retreating glaciers and any remaining buffering against precipitation deficits needs to come from other stores, e.g. the snowpack and groundwater.

For comparison of the effect of the two threshold approaches on analysing drought processes, a dynamic glacier area conceptualisation was used for both thresholds (Fig. 10). The different thresholds clearly result in the identification of contrasting streamflow droughts in the Wolverine catchment in 2091. The HVT shows a long drought from July until October (shortly interrupted in September), while the TVT shows many streamflow droughts during the whole year (Fig. 10). The glacier has

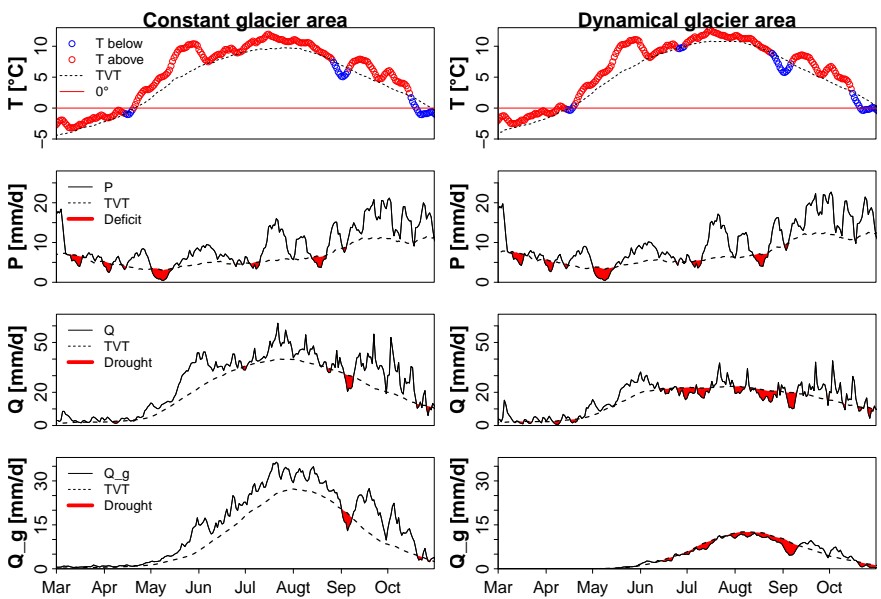

**Figure 9.** Example of streamflow droughts and causing factors (T, P and Q_g) for the different glacier area conceptualisations. Multi-model mean temperature, precipitation and discharge time series are presented for the Nigardsbreen catchment for March-October 2092, based on climate scenario RCP 8.5. For the time series of P and T a 7-day moving average was used. Droughts are analysed with the transient threshold. Note that T and P are slightly different in the left and right panels due to different lapse rates obtained during the calibration.

disappeared in 2091 in the Wolverine catchment, which caused a change in the regime. The HVT is based on the historical regime and the 'drought' that can be seen is essentially the mismatch between the old and new regime. Therefore, this drought occurs every year at the same moment, since the HVT is not changing and there is no glacier any more to produce a discharge peak in the summer. This 'drought' does not represent extreme or exceptional discharge values and relating it to anomalies in P and T is not possible. T anomalies are mostly above the HVT temperature threshold, due to a warming climate and can therefore not directly be used as explanation for droughts. Also the deficits in P can not explain the large drought in the discharge. However, in the TVT approach, the threshold has adapted to the reduced summer discharge, like the thresholds of P and T have adapted (Fig. 10 right panel). This causes temperatures to fluctuate around the threshold and these anomalies can be used to analyse the causing factors of drought in Q. Also the deficits in P can be related to the droughts that are occurring in the streamflow. The TVT approach therefore could be used to study which drought processes and drought types (Van Loon and Van Lanen, 2012) will become important in the future.

## 5   Discussion

In this study we aimed to systematically test the role of glacier dynamics and threshold approaches in simulating and analysing future streamflow droughts in glacierised catchments. The results indicate different effects of both methodological choices on

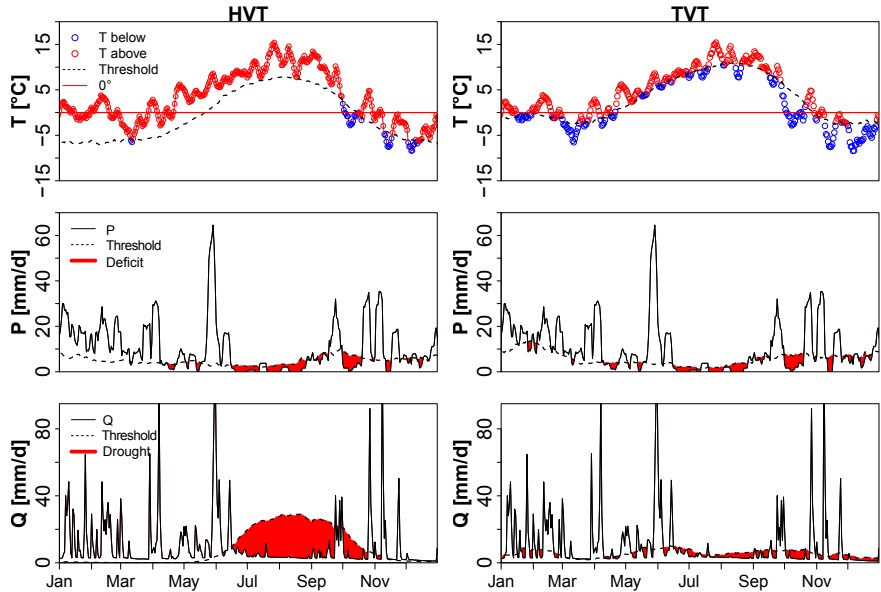

**Figure 10.** Example of streamflow drought and causing factors (T and P) for the different threshold methods. Temperature, precipitation and discharge time series are presented for the Wolverine catchment for 2091, based on climate scenario RCP 4.5 and the dynamic glacier area conceptualisation. For the P and T time series a 7-day moving average was used.

drought characteristics and the analysis of drought processes, which is of major importance for further studies analysing climate change effects on streamflow droughts in cold climates. The study also showed that the methodological choices highlight different aspects of future streamflow droughts and it is therefore essential for further studies to determine which aspect of drought one wants to study and choose the methods accordingly.

As glaciers have been shrinking and likely will further shrink in the future (e.g. Vaughan et al., 2013), there is wide consensus that glacier change needs to be accounted for in hydrological modelling. However, we have shown in this study that modelling with a constant glacier area can be interesting to analyse seasonal drought processes in the future, without taking into account the long term changes of the glacier area. Analysing drought processes usually includes looking at anomalies in precipitation and temperature and their propagation through the hydrological cycle. Most drought processes occur within the

season (Van Loon and Van Lanen, 2012) but some drought types can be classified as multi-season drought. An example is the snowmelt drought, which can be caused by high temperatures or low precipitation in winter, resulting in less snow supply to the snowpack, causing a drought in the snowmelt peak in summer due to less snow available for melt (Van Loon et al., 2015). In glacierised catchments the time between the meteorological drivers and the resulting drought in streamflow can be even longer due to the long response time of glaciers (Bahr et al., 1998; Roe and O'Neal, 2009). A reduced winter mass balance would not

directly result in a streamflow drought in the glacier melt peak if temperatures are above or close to normal in summer. However, after several negative mass balance years and consequent glacier retreat, less glacier area and volume will be available for

melt-water generation, possibly resulting in a drought when temperatures are close to or below normal in summer. Thus, the long term effects of dynamical glaciers can influence droughts. Separating the effects of short term climate variability and a changing glacier area and volume on droughts by using a constant and dynamic glacier area can therefore give useful insights on these intertwined processes.

Another option regarding the glacier modelling could be the full removal of the glacier. In theory, the comparison of simulated discharge without glaciers, with constant glaciers and with dynamic glaciers can give interesting information about the role of glaciers in causing or preventing streamflow droughts. For example, apart from distinguishing between the anomalies in glacier melt and glacier dynamics as causing factors of streamflow drought, also anomalies in snow melt and precipitation deficits in relation to streamflow droughts could be better assessed. However, model parameters are calibrated to discharges

and glacier mass balances of glacierised catchments and therefore reflect the typical sensitivities and relations among fluxes for glacierised catchments. Hence, these parameters cannot be directly used to simulate a non-glacierised catchment. We therefore did not include this option explicitly in our study. Nevertheless, in our dynamic glacier conceptualisation we simulate a glacier disappearance for the Wolverine catchment from around 2060 onwards, while still using the same parameters. A solution, however, with time-varying parameters for simulation of long time periods and retreated glaciers does not yet exist (see e.g.

Merz et al., 2011; Thirel et al., 2015; Heuvelmans et al., 2004; Paul et al., 2007; Farinotti et al., 2012).

    The dynamic glacier area representation used in this study is a simplification and therefore has its limitations. The $\triangle$h-parametrisation in HBV-light can for example not be used to simulate glacier advance compared to the defined glacier profile (see also Huss et al., 2008, 2010). Moreover, Huss et al. (2008) mention that this parametrisation is not able to reproduce the time scales for transfer of mass from the accumulation area to the ablation area. The change in volume is distributed over the

glacier area to simulate an elevation change at the end of each year. Response time effects on drought can therefore not be directly analysed. However, the constant and dynamic glacier area conceptualisations are able to show the effect of short term climate variability and long term glacier area changes on streamflow droughts. Another drawback, in this HBV-light model version, is that elevations do not change after melting of glaciated model units. The surface lowering may in reality result in a positive feedback of melt due to higher temperatures and potentially less precipitation. Furthermore, this model version

does not allow to use a seasonally varying discharge as benchmark in the calibration (instead of the mean discharge, see eq. 1), which would be preferred when the regime shows a strong seasonality (Schaefli and Gupta, 2007). However, our objective function is not based on the whole discharge time series, but only on the seasonal and peak discharges and the glacier mass balances, thereby partly taken the problem of calibrating on the mean discharge into account.

    Despite these limitations the implementation of the dynamic glacier area in the HBV model is an important improvement for

the hydrological modelling in glacierised catchments. Many of the global hydrological models that have so far been applied to estimate changes in streamflow drought have not included glacier dynamics or any glacier component at all (e.g. Zhang et al., 2013). Compared to catchment scale hydrological models which use approaches where glacier area is adjusted in larger jumps, without the coupling between melt and ice volume, (e.g. Juen et al., 2007) the dynamic glacier area method used here is more applicable for the transient drought threshold approach because of the gradually changing discharge regime due to the

gradually changing glacier. Using more advanced models to simulate glacier retreat may result in slightly different numbers in

the timing of glacier retreat and changes in the discharge regime, but it would not change the results of this study regarding the use of the methodological options for drought analysis.

In our study, the glacier disappearance simulated by 2060 for the Wolverine catchment might be an unrealistically extreme result for most of the glacierised catchments in the world (Zemp et al., 2006; Rees and Collins, 2006; Radić et al., 2014; Bliss et al., 2014; Huss and Hock, 2015). The use of a calibrated conceptual glacio-hydrological model in our study which uses a simplification of glacier processes and does not take into account e.g. a varying lapse rate (Gardner and Sharp, 2009), firn on the glacier, reduced albedo due to melt and explicit en- and subglacial drainage, might have influenced the glacier melt and thereby also the rate of glacier disappearance. Also the absence of a snow redistribution routine in our model, in which snow from higher elevation zones can be redistributed to the glacier (Seibert et al., 2017), might have influenced the rate of glacier retreat. The snow towers that appeared in our model, because snow was not redistributed (see also Freudiger et al., 2017), were checked for their possible error on the discharge simulations. The amount of SWE stored (or released in some elevation zones in the future) in the snow towers compared to the total discharge was however small (negligible up to a few percent). We therefore considered the effect of snow towers on our drought analysis to be small. Also the assumption that parameters stay constant over time, while the catchment and climate are changing (Merz et al., 2011) (in this case changing glaciers) is causing some uncertainty.

We should also keep in mind that the future glacier area evolution has a large uncertainty caused by climate model uncertainties as shown in this study for the Nigardsbreen catchment (Fig. 5). The historical glacier area changes for Wolverine agree with the observed glacier area at the end of the historical period, but for Nigardsbreen a smaller glacier area than observed is simulated. This could be caused by the simplified modelling of glacier processes, the construction of the glacier profile and/or due to the climate forcing. We compared the annual average glacier melt contribution in Nigardsbreen catchment with Engelhardt et al. (2014) and find comparable results (around 20%). Nevertheless, both uncertainties, in the model and forcing, mainly influence the timing of changes in both catchments but not the processes that we studied and compared in the different scenarios, which is the main focus of this study.

Moreover, the two case study catchments in this study, with a different glacier area evolution and resulting changing discharge regime, showed the range of possible effects the methodological choices can have on future streamflow and drought projections. The glacier disappearance in the Wolverine catchment is a highly relevant and clear example in the discussion about drought definitions and thresholds in future projections. It also illustrates that the hydrological regime becomes more variable when the catchment changes from highly-glacierised to non-glacierised (Fountain and Tangborn, 1985). This is important for streamflow drought analysis, since streamflow droughts will be more variable and mainly dependent on variability in precipitation and it is therefore not appropriate to use a historical threshold that is based on other hydrological processes (stable glacier-dominated regime).

The other choice, which threshold approach to use, mainly relates to the question of the definition of a drought. For streamflow drought projections a comparison with a historical period is always needed in order to assess the changes and to be able to understand them. However, one can raise the question if the threshold needs to be the same in the two periods (HVT approach). The results showed that due to the regime shift the HVT indicates severe droughts every year in summer. If we would have

applied pooling (Fleig et al., 2006), the differences in drought characteristics between the threshold methods due to the regime shift would have been even more pronounced. Because this 'regime shift drought' occurs each year it will become the normal situation and it is clear that this mismatch of regimes can not be regarded as a drought. Therefore, the transient threshold is a better option to study droughts in glacierised catchments where discharge regimes change. Moreover, the advantage of TVT is

that it can be used to analyse future drought processes which will be an important aspect for future water management. This study agrees with the findings of Wanders et al. (2015) that different threshold approaches can have substantial effects on future streamflow drought characteristics. Furthermore the results confirm the findings of Van Huijgevoort et al. (2014) and Wanders et al. (2015) that in cold climates where regime shifts are expected the TVT is a better identifier of droughts than HVT. This is especially the case in glacierised basins as shown in this study, which are rapidly changing due to glacier retreat.

However, using the TVT, changes between historical and future situations cannot be assessed, because the benchmark itself is changing. Most studies (e.g. Forzieri et al., 2014) looking at future droughts in low flow periods have used a historical threshold to define future droughts and conclude that low flows will increase and therefore less droughts will occur. Here, the normal situation is changed (higher low flows), which is identified using the HVT. This information about changing normals is lost when only drought characteristics are analysed using the TVT. It is therefore important to complement the TVT drought

characteristics with an analysis of the changes in the regime to put the drought results into perspective. This could be done for example by looking at the changes in the TVT itself or comparing the TVT with the HVT, and by checking annual discharges (Figs. 7 and 6). In this study the annual discharges of the Wolverine catchment are decreasing in the future, whereas the signal for Nigardsbreen is less clear. Apart from a changing seasonality, these annual discharges give information on how the total water availability will change.

Both threshold approaches thus take another viewpoint of drought. With the HVT we look at future droughts from a viewpoint now and with the transient threshold we change our viewpoint to the future and we then look at droughts. Since future droughts will also have impacts in the future, the latter viewpoint is more logical to study future droughts. However, the TVT also has some uncertainties. The main uncertainty concerns the adaptation that is assumed when using the transient threshold. The transient threshold changes every year and not always in the same direction and with the same magnitude. This would

mean that society and ecosystems need to be flexible in the adaptation and the question is how adaptable we are to these regime changes and if we can assume that a same level of adaptation can be reached in both climate change scenarios (RCP 4.5 and RCP 8.5). Vidal et al. (2012) for example discuss in their study about future droughts in France, in which the baseline of a standardized drought index is adapted each month, the feasibility of this time step and compare it with adaptation time scales for irrigated crops (seasonal or annual) and forestry (decadal). Nevertheless, several studies argue the use of 'constant normals'

as being representative for both the current and future climate and indicate ways to derive changing normals (e.g. Livezey et al., 2007; Arguez and Vose, 2011; Vidal et al., 2012).

Another aspect of the discussion about the definition of a drought is the use of a variable threshold to identify droughts. In contrast to other studies which specifically look at low flow periods to analyse droughts (see e.g. Hisdal et al., 2001; Fleig et al., 2006; Feyen and Dankers, 2009; Forzieri et al., 2014), for example by using a constant instead of daily varying threshold, we

include streamflow deficiencies in the high flow season as well in our streamflow drought definition. This is also done in many

other studies that use a variable threshold level method (e.g. Van Loon et al., 2015; Fundel et al., 2013) or standardised drought indices (e.g. Shukla and Wood, 2008; Vidal et al., 2010), or in global scale future drought studies (e.g. Van Huijgevoort et al., 2014; Prudhomme et al., 2014; Wanders et al., 2015), because it does fit with the definition of drought as below normal water availability (Tallaksen and Van Lanen, 2004). However, the spatial and temporal scales in these studies can be different from our scales. Consequently, not all our identified streamflow droughts will lead to impacts. Nonetheless, in general these droughts in terms of streamflow deficiencies might be important for, and could impact, downstream water users. It would be interesting to apply the methods and outcomes of this study to other glacierised catchments around the world, in particular those which are drier and therefore more dependent on glacial meltwater (e.g. Gascoin et al., 2011) and where climate change will likely have impacts on water availability and droughts.

## 6   Conclusions

This study systematically elucidated the effect of glacier dynamics and threshold approach on future streamflow drought characterisation and the analysis of the governing hydrological processes. The discharges and streamflow droughts of two case study catchments, Nigardsbreen (Norway) and Wolverine (Alaska), with a currently high percentage of glacier cover were studied. Streamflow was modelled with the HBV-light model for a historical period and into the future. This model accounts for the glacier retreat but also allows to keep glaciers constant, a feature that enabled this study to carry out a comparison of four potential views on future streamflow droughts. Assuming a constant glacier area and a threshold approach whereby droughts are defined based on the historical hydrological regime, results in almost no droughts in the future, due to an increase in glacier melt. When the same historical threshold approach is applied to discharge simulated with glacier change, results show severe 'regime shift droughts' in summer due to retreat, or even complete disappearance (Wolverine), of the glacier. If future droughts are studied from a future perspective, by using a transient threshold that changes with the changing hydrological regime, differences in drought characteristics between historical and future periods, and glacier dynamics options are smaller. Drought characteristics greatly differ among the four scenarios and these choices will therefore strongly influence future drought projections. We found the four options to be able to answer different questions about future streamflow drought in glacierised catchments: the transient threshold for analysing drought processes in the future, the historical threshold approach to assess changes between historical and future periods, the constant glacier area conceptualisation to analyse the effect of short term climate variability and the dynamic glacier area to model realistic future discharges in glacierised catchments.

Most important for further future streamflow drought studies is to define what a future drought is and subsequently choose the right method. In addition to the definition of future droughts, questions that also need to be addressed in further studies is the relation between the statistical description of droughts (the threshold based on a percentile of the flow duration curve) and the impacts and experiences of droughts by ecosystems and society. Are all droughts detected in the high flow season also experienced as droughts, or for example only droughts with high deficits or long durations? Streamflow droughts upstream would mainly impact energy production and river ecology. However, if for example enough reservoir capacity is present for the energy production, a deficit in a part of the melt peak might be compensated by higher discharges from the glacier during

the rest of the melt season and no impact is felt. In this study an upstream perspective was used, but many people depending on the water from glaciers live more downstream (e.g. water dependency in the Himalayas). Streamflow droughts in the high-flow season upstream in glacierised catchments are related to droughts in the low-flow season downstream, with potentially even larger impacts. Further research should investigate this relation and the impacts of drought downstream in these regions.

## 7  Data availability

Data for the Nigardsbreen catchment are available via the Norwegian Water Resources and Energy Directorate (NVE) and for the Wolverine catchment via U.S. Geological Survey (USGS). Streamflow data and mass balances for Wolverine are also available online (https://waterdata.usgs.gov/nwi and https://alaska.usgs.gov/products/data.php?dataid=79). Ice thickness maps are available via Matthias Huss and for some Norwegian glaciers via NVE. The climate model data are available from the Coordinated Regional Climate Downscaling Experiment (CORDEX) (http://www.cordex.org/). Glacier outlines can be obtained from GLIMS and NVE (Nigardsbreen) (http://www.glims.org/RGI/rgi50_dl.html and https://www.nve.no/hydrologi/bre/bredata/). The ASTER DEM can be downloaded from http://reverb.echo.nasa.gov/reverb/ and ERA-interim data from http://apps.ecmwf.int/datasets/.

### Appendix A:  Model parameters glacier and snow routine

In Table A1 the calibrated parameter values that were used in the glacier and snow routine of the HBV-light model are presented. A different parameter set was obtained for the dynamic and constant glacier area conceptualisations. The refreezing coefficient ($CFR$), which determines the amount of refreezing liquid water in the within the snowpack when temperatures are below the threshold temperature, and the water holding capacity of snow ($CWH$), which determines how much meltwater and rainfall are retained within the snowpack, were assigned a constant value and not calibrated. $KG_{min}$, $dKG$ and $AG$ are the parameters for the glacial water storage-outflow relationship (Stahl et al., 2008). The degree-day factor ($CFMAX$) is multiplied with $CF_{glacier}$ to simulate glacier melt and it is multiplied (divided) by $CF_{slope}$ to calculate melt of snow and ice for south-facing slopes (north-facing slopes). No correction is used for east- and west-facing slopes.

**Table A1.** Glacier and snow routine parameter values. All parameters were calibrated except $CFR$ and $CWH$, indicated with *. For each catchment two parameter sets were obtained, one for the dynamical glacier conceptualisation (D) and one for the static glacier area conceptualisation (C).

| Parameter | Description | Nigardsbreen - C | Nigardsbreen - D | Wolverine - C | Wolverine - D |
|---|---|---|---|---|---|
| $T_{calt}$ [°C/100 m] | T lapse rate | 0.65 | 0.55 | 0.54 | 0.46 |
| $P_{calt}$ [%/100 m] | P lapse rate | 13.40 | 15.43 | 15.98 | 12.70 |
| $TT$ [°C] | Threshold temperature | $-0.17$ | $-0.32$ | 0.04 | 0.12 |
| $CFMAX$ [mm/d°C] | Degree day factor | 2.34 | 3.17 | 2.67 | 1.94 |
| $SFCF$ [-] | Snowfall correction factor | 1.00 | 0.95 | 1.69 | 1.88 |
| $CFR$ * [-] | Refreezing coefficient | 0.05 | 0.05 | 0.05 | 0.05 |
| $CWH$ * [-] | Water holding capacity of snow | 0.1 | 0.1 | 0.1 | 0.1 |
| $CF_{glacier}$ [-] | Glacier melt correction factor | 1.32 | 1.18 | 1.80 | 1.72 |
| $CF_{slope}$ [-] | Slope melt correction factor | 2.67 | 1.54 | 1.65 | 2.57 |
| $KG_{min}$ [1/d] | Minimum outflow coefficient glacier storage | 0.20 | 0.20 | 0.20 | 0.20 |
| $dKG$ [1/d] | Maximum minus minimum glacier storage outflow coefficient | 0.50 | 0.39 | 0.50 | 0.50 |
| $AG$ [mm] | Calibration parameter | 0.003 | 1.25 | 9.95 | 0.0003 |

*Author contributions.* AVL and MVT conceived and designed the study. MVT carried out the modelling and analyses with feedback from AVL and MV. NW and MV provided the R-script for TVT calculation and the HBV model with glacier routine, respectively. AVL and AT supervised the MSc thesis work that formed the basis of this paper. MVT, AVL and AT discussed the structure and content of the manuscript. MVT wrote the manuscript. All co-authors edited and revised the manuscript and approved the final version (AVL, NW, KS, MV, AT).

5   *Competing interests.* The authors declare that they have no conflict of interest

*Acknowledgements.* We would like to thank Shad O'Neel and Louiss Sass from USGS for providing the mass balance data and climate observations for the Wolverine catchment. We thank Wai Kwok Wong, Liss Andreassen and Kjetil Melvold from NVE for providing the ice thickness data, the mass balance data and metadata and the climate and discharge observations for the Nigardsbreen catchment. We also thank Annemiek Stegehuis for extracting the CORDEX climate model data for the two catchments and Matthias Huss for providing the

10   Wolverine ice thickness map. Furthermore, we thank Nick Barrand (University of Birmingham) for his support when setting up this research. We thank the University of Birmingham for covering the article processing charge. Finally, we thank the three reviewers, the editor and L.M. Tallaksen for their helpful comments. AVL is funded by NWO Rubicon project no. 2004/08338/ALW and NW is funded by NWO Rubicon 825.15.003. This paper was developed within the framework of the UNESCO-IHP VIII FRIEND programme (EURO-FRIEND - Low flow and Drought group).

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
