# Peer review of "The role of glacier changes and threshold definition in the characterisation of future streamflow droughts in glacierised catchments"

_Hydrology and Earth System Sciences, 2017_

## Referee Comment (RC1) · Anonymous Referee #1 · 11 May 2017

Review of The role of glacier dynamics and threshold definition in the characterization of future streamflow droughts in glacierised catchments By M. Van Tiel et al. Submitted to Hydrology and Earth System Sciences General comment This paper is an analysis on the possible causes of streamflow droughts in glacierised catchments in the context of climate change (affecting glacier geometry, melt rate, discharge regime and drought). The authors have chosen 2 different highly (more than 60%) glacierised catchments for their study. They considered 2 different approaches for modeling glacier change: (1) glacier topography remains constant, and (2) glacier topography is empirically updated every year according to surface mass balance. First, different approaches on how taking into account changes in glacier geometry in streamflow modeling is discussed. Since the approaches used are empirical, no ice dynamics in the strict sense is considered. So I suggest to modify ''glacier dynamics'' into ''glacier changes'' in the title. Since glacier topography always changes in a changing climate, it is in my opinion not a sound option to analyze streamflow evolution without adapting the glacier topography accordingly. A more realistic option would be to assume no glacier at all. But assuming a constant and arbitrary glacier surface area in a changing climate will produce streamflow results which can hardly be interpreted. Furthermore, I feel that the different ways of defining a drought a bit confusing. Since I am not hydrologist, I apologize for that. But in my opinion, it would be sufficient to define a reference period (RP) (f. i. 1960-1990 as used in Switzerland for climatology), and to present streamflow results outside this RP as deviations from it. I think results presented this way will be more useful for water management purposes. The paper is well written and the results well presented. I can recommend publication.

Specific comments: p. 18 line 14: ''. . . in de left . . .'' needs correction

Please also note the supplement to this comment:
http://www.hydrol-earth-syst-sci-discuss.net/hess-2017-119/hess-2017-119-RC1-supplement.pdf

---

## Referee Comment (RC2) · M. Zappa (Referee) · 23 May 2017

[referee-annotated manuscript omitted]

---

## Author Comment (AC1) · 24 May 2017

**Response to referee comment Anonymous Referee #1**

We would like to thank Anonymous Referee #1 for reviewing our manuscript and the positive feedback and the suggestions for improvement. We will reply to the comments below.

The reviewer's comments are in **bold**, our response in *italic.*

**General comment**
**This paper is an analysis on the possible causes of streamflow droughts in glacierised catchments in the context of climate change (affecting glacier geometry, melt rate, discharge regime and drought). The authors have chosen 2 different highly (more than 60%) glacierised catchments for their study. They considered 2 different approaches for modeling glacier change: (1) glacier topography remains constant, and (2) glacier topography is empirically updated every year according to surface mass balance.**

**First, different approaches on how taking into account changes in glacier geometry in streamflow modeling is discussed. Since the approaches used are empirical, no ice dynamics in the strict sense is considered. So I suggest to modify ''glacier dynamics'' into ''glacier changes'' in the title.**

*>> The approaches to take into account changes in glacier geometry are indeed empirical and complex ice flow modelling is not used. We used the term dynamics to indicate that the glacier geometry change and streamflow modelling is coupled and therefore the glacier is adjusted in a dynamical way. However we agree that this may be confusing and we can change the "glacier dynamics" into "glacier changes" in the title.*

**Since glacier topography always changes in a changing climate, it is in my opinion not a sound option to analyze streamflow evolution without adapting the glacier topography accordingly. A more realistic option would be to assume no glacier at all. But assuming a constant and arbitrary glacier surface area in a changing climate will produce streamflow results which can hardly be interpreted.**

*>> We agree that glacier geometry (in the model described by glacier area and glacier thickness in each elevation zone) always changes in a changing climate and that one should account for that in the modelling of glacierised catchments to obtain realistic predictions, as we discuss in the manuscript. The benchmark simulation with static glaciers in our study can still serve a number of interesting purposes. Most importantly, we can use the static glacier modelling to isolate the direct effect of changes in temperature and precipitation from the effect changes in glacier extent, so that we can understand the corresponding changes in seasonal streamflow variability better. While the resulting streamflow quantities might not be realistic the benchmark can still be useful for increasing our understanding of how streamflow hydrograph changes will lead to changes in the processes leading to streamflow droughts. This benchmark method has also been used many times in hydrological modelling studies (see e.g. Akhtar et al., 2008; Stahl et al., 2008; Tecklenburg et al., 2012; Sun et al., 2015. Finally, some models still only use a constant glacier cover through time, either since they only model a short period of time or since the model does not allow to model glacier change (see e.g. Singh & Kumar, 1997; Klok et al., 2001; Shabalova et al., 2003; Verbunt et al., 2003; Singh & Bengtsson, 2005; Terink et al., 2005; Horton et al., 2006; Rahman et al., 2013). We will make these points clearer in the revised manuscript.*

*Assuming no glacier is certainly a possible option; yet a less defendable one as the model parameters were calibrated to glacierised catchments and model parameters thus effective will reflect the typical sensitivities and relations among fluxes of glacierised catchments. Nevertheless, in our simulations for the Wolverine catchment a no glacier scenario also occurred during part of the simulation period and can happen in general in studies of future simulations of glacierised catchtments. However, no*

*solution to avoid this problem with calibrated model parameters and changing glaciers and long simulation periods does exist. We will improve the discussion on this point.*

**Furthermore, I feel that the different ways of defining a drought a bit confusing. Since I am not hydrologist, I apologize for that. But in my opinion, it would be sufficient to define a reference period (RP) (f. i. 1960-1990 as used in Switzerland for climatology), and to present streamflow results outside this RP as deviations from it. I think results presented this way will be more useful for water management purposes.**

*>> We used and compared two methods to define streamflow droughts in our study: the Historical Variable Threshold (HVT) and Transient Variable Threshold (TVT). For the HVT we use a fixed RP period in the past, as the reviewer describes, and deviations from this Historical Threshold in the future (below threshold values) is what we define as streamflow droughts, in the HVT method. The aim of this paper was to compare this approach with another threshold approach, the TVT. This Transient Threshold changes according to the changing regime because there is no fixed reference period (it is a moving 30-year window). The results show that using a HVT or RP in changing catchments, like glacierised catchments, may not always be the most useful method for water management purposes since this method indicates changes in the regimes but not real changes in streamflow droughts. In our study, in the Nigardsbreen catchment for example, the HVT is based on a period where the catchment was highly glacierised and using this threshold outside the reference period (in the future) where the glacier has retreated gives a large increase in drought deficit while in fact it is only a small shift in timing. The HVT method thus also leads to a strange comparison of different streamflow generation processes controlling the streamflow signal and variability (especially in the Wolverine catchment where the glacier has disappeared). We do agree that it is important to look at changes and deviations from a past RP, e.g. by looking at changes in the hydrological regime (see Fig. 5), but for water management we think it is also relevant to take these changes and adaptation into account and change the point of view and look at future streamflow variability to analyse streamflow droughts. We will clarify our reasoning for using and comparing the different thresholds in the revised manuscript.*

**The paper is well written and the results well presented.**

**I can recommend publication.**

*>> Thank you for this positive evaluation.*

**Specific comments:**
**p. 18 line 14: "… in de left …'' needs correction**

*>> Thank you, we will correct this in the revised version.*

References:

Akhtar, M., Ahmad, N., & Booij, M. J. (2008). The impact of climate change on the water resources of Hindukush–Karakorum–Himalaya region under different glacier coverage scenarios. *Journal of hydrology*, *355*(1), 148-163.

Horton, P., Schaefli, B., Mezghani, A., Hingray, B., & Musy, A. (2006). Assessment of climate-change impacts on alpine discharge regimes with climate model uncertainty. *Hydrological Processes*, *20*(10), 2091-2109.

Klok, E. J., Jasper, K., Roelofsma, K. P., Gurtz, J., & Badoux, A. (2001). Distributed hydrological modelling of a heavily glaciated Alpine river basin. *Hydrological Sciences Journal*, *46*(4), 553-570.

Rahman, K., Maringanti, C., Beniston, M., Widmer, F., Abbaspour, K., & Lehmann, A. (2013). Streamflow modeling in a highly managed mountainous glacier watershed using SWAT: the Upper Rhone River watershed case in Switzerland. *Water resources management*, *27*(2), 323-339.

Shabalova, M. V., Van Deursen, W. P. A., & Buishand, T. A. (2003). Assessing future discharge of the river Rhine using regional climate model integrations and a hydrological model. *Climate Research*, *23*(3), 233-246.

Singh, P., & Kumar, N. (1997). Impact assessment of climate change on the hydrological response of a snow and glacier melt runoff dominated Himalayan river. *Journal of Hydrology*, *193*(1), 316-350.

Singh, P., & Bengtsson, L. (2005). Impact of warmer climate on melt and evaporation for the rainfed, snowfed and glacierfed basins in the Himalayan region. *Journal of Hydrology*, *300*(1), 140-154.

Stahl, K., Moore, R. D., Shea, J. M., Hutchinson, D., & Cannon, A. J. (2008). Coupled modelling of glacier and streamflow response to future climate scenarios. *Water Resources Research*, *44*(2).

Sun, M., Li, Z., Yao, X., Zhang, M., & Jin, S. (2015). Modeling the hydrological response to climate change in a glacierized high mountain region, northwest China. *Journal of Glaciology*, *61*(225), 127-136.

Tecklenburg, C., Francke, T., Kormann, C., & Bronstert, A. (2012). Modeling of water balance response to an extreme future scenario in the Otztal catchment, Austria. *Advances in Geosciences*, *32*, 63.

Terink, W., Lutz, A. F., Simons, G. W. H., Immerzeel, W. W., & Droogers, P. (2015). SPHY v2. 0: Spatial Processes in HYdrology. *Geoscientific Model Development*, *8*(7), 2009-2034.

Verbunt, M., Gurtz, J., Jasper, K., Lang, H., Warmerdam, P., & Zappa, M. (2003). The hydrological role of snow and glaciers in alpine river basins and their distributed modeling. *Journal of hydrology*, *282*(1), 36-55.

---

## Short Comment (SC1) · 2 Jun 2017

The paper addresses an important topic related to the influence of glaciers on the flow regime in a future (warmer) climate, and drought in particular. My remarks relate primarily to the terminology used for defining drought and do not address the full paper as such.

Two different threshold approaches are employed; a threshold based on the historical period and a transient threshold approach, whereby the threshold adapts every year in the future to the changing regimes. In both cases, drought occurs when the discharge falls below the threshold. A daily variable threshold is used (80th percentile), defined

based on a 30-day moving average time series. There is no seasonal distinction made and droughts can occur any time of the year as long as the flow is below the daily varying threshold.

The study, which is based on two catchments, projects "extreme increases in drought severity in the future" for the scenario HVT-D, i.e. a historical threshold combined with a dynamical glacier area. More specifically, the simulations show a lower peak flow and a shift towards an earlier melt peak, implying higher than normal flow early in the summer season and lower than normal flow towards the end of the melt period (ref. Figure 7). Accordingly, the projected increase in drought severity (from the time of the peak and onwards) is mainly caused by a change in the timing of the melt peak, or as stated in the paper, "by the regime shift due to a reduction in glacier area". (It is recommended to use the same scale on the y-axis for the different plots in Figure 7 to ease the comparison.)

Both catchments have typically glacier flow regimes with low flows in winter and high flows in summer. Projected changes in flow seasonality in catchments with glaciers are strongly linked to changes in the snow regime with more precipitation falling as rain (rather than snow) and less snow accumulating (with the exception of some high altitude regions). Milder winters are projected to lead to earlier spring flood, a tendency that can already be observed for Norway (Wilson et al., 2010). Similar, warmer spring and summers are projected to lead to earlier and more glacier melt (as long as the glacier volume does not reduce too much). However, a shift in the timing or a reduction in the flow during the snow or glacier melt season is not associated with an increase in drought in these cold climate regions; neither by the snow/glacier research communities nor by water management. Rather, if focus is on drought, there is a concern that a longer snow free season combined with an increase in evapotranspiration may lead to increased drought in the following low flow period (e.g. Wilson et al., 2010). Glacierised catchments located in wet climates such as western part of Norway are further expected to be less prone to droughts in the future as compared to catchments

located in drier climates.

The terms 'flood' and 'drought', as well as 'high flow' and 'low flow' periods are well defined concepts in hydrology, and I would strongly argue against using the term 'drought' for a period with relatively low flow during the high flow season or equivalent, 'flood' for a period with relatively high flow during the low flow season, merely based on their percentage deviations from the seasonal flow regime (and not their impacts). Rather, I suggest referring to these deviations as streamflow anomalies (or deficiencies for drought) as originally proposed by Stahl (2001) when introducing the daily varying threshold approach, and later elaborated in Hisdal et al. (2004). As highlighted in these studies, the variable threshold approach is adapted to detect streamflow deviations during both high and low flow seasons, and periods with relatively low flow during the high flow season are commonly not considered droughts. Still, lower than normal flows during high flow seasons may be important for later drought development.

References

Hisdal, H., Tallaksen, L.M. Clausen, B. Peters, E. & Gustard, A. (2004) Hydrological Drought Characteristics. In: L.M. Tallaksen & H.A.J. van Lanen (Eds), Hydrological Drought Processes and Estimation Methods for Streamflow and Groundwater. Developments in Water Sciences 48. Elsevier B.V., the Netherlands, 139-198.

Stahl, K. (2001). Hydrological drought: A study across Europe (Doctoral dissertation, Institut für Hydrologie der Universität). Available online through: freidok.uni-freiburg.de

Wilson, D., Hisdal, H and Lawrence, D. (2010) Has streamflow changed in the Nordic countries? – Recent trends and comparison to hydrological projections. J. Hydrol., 394, 334-346.

---

## Author Comment (AC2) · 4 Jul 2017

**Response to referee comment Massimiliano Zappa**

We would like to thank Massimiliano Zappa for reviewing our manuscript and the feedback and the useful suggestions for improvement. We will reply to the comments below.

The reviewer's comments are in **bold**, our response in *italic.*

**Dear authors,**

**I like the manuscript but I have a couple of concerns:**

**a) Glacier dynamics: C D: You are comparing a state-of-the-art approach (dynamical glaciers), with an approach which is acknowledged to be inadequate in transient mode (constant glaciers). I think you should have a look at the option without glacier change, but tax it as inadequate early enough in the paper (Figure 6) and continue with the dynamical model only. Up to Figure 6 you could also work with a version where you remove the glaciers from the beginning.**

*>> We agree that constant glaciers are not realistic if used for prediction and discuss this in the manuscript. As the reviewer notes, it is interesting to have a look at this option, since not all hydrological models (at all scales) do yet use a dynamical glacier approach and we want to show the effect of this glacier modelling choice on streamflow drought analysis. In the revised manuscript, we will make clearer from the beginning that constant glaciers are not realistic in transient mode. However, we do think this setting is useful and we therefore do not agree that we should only continue with dynamical glacier modelling from Figure 6 onwards. From Figure 6 onwards, especially in Figure 8, we show that modelling with constant glaciers is actually an interesting benchmark simulation. We can use it to isolate the effect of short term anomalies in precipitation and temperature from the effect of long term glacier changes on streamflow droughts and it thus gives more insight in the processes causing streamflow droughts in glacierised catchments (see also comment to reviewer #1). In relation to this, a version where we would remove the glaciers from the beginning would be a very interesting option, because it would give more insight to what extent streamflow droughts are caused by precipitation anomalies and/or snow melt anomalies (no glacier version), glacier melt anomalies (constant glacier version) and glacier dynamics (dynamical glacier option). However, as mentioned in the reply to reviewer #1, model parameters are calibrated to glacierised catchments and therefore reflect the typical sensitivities and relations among fluxes of glacierised catchments and cannot directly be used to simulate catchments without a glacier.*

**b) HVT TVT: Here I fear that regime shift (shown by HVT) and actual drought-analysis (shown by TVT) are mixed up. In the discussion part this is presented as finding, but I think this should be stated from the beginning.**

*>> Interestingly, reviewer #1 had the opposite suggestion, of using only a HVT approach for future drought analysis. We mention in the introduction that using a HVT leads to severe 'droughts' in case of regime shifts (Van Huijgevoort et al., 2014). Despite this, HVT is used in studies to analyse future streamflow droughts at the global scale, which we also mention in the introduction. In the threshold level method droughts are defined as discharges below the threshold, whether HVT or TVT, so purely looking at the definition both are droughts. However, we do discuss this issue of regime shifts and whether regime-shift 'droughts' should also be interpreted as droughts. Given the confusion of both reviewers and the lack of consideration of this drought definition issue in most global future drought studies we infer that it is useful to discuss the effects of certain methods (HVT vs. TVT). We conclude (with the reviewer) that some are better suitable for the drought analysis than others, depending on the focus of a study/what the research interest is. We will clarify this in the revised version (in the discussion).*

**c) For calibration and validation I would use a seasonally varying discharge value instead of the average discharge as benchmark (see detailed comment in the commented manuscript)**

*>> This is a good point. We agree that Nash Sutcliffe criterion is not the best objective function to use in areas with strong seasonal discharge. It would indeed be better to use the monthly mean of the observations. However, such an objective function is not available (yet) within the calibration tool of the HBV-light model. We will discuss this issue in the discussion. Also, the objective function that we used is not based on the whole discharge time series, but for 40% on glacier mass balances, 40% on seasonal discharge and 20% on the peak flows. We therefore do not calibrate on the whole seasonal cycle and avoid this problem to a certain extent. But still, using the monthly mean of the observations in the seasonal discharge calibration part would be beneficial.*

**Please clarify these issues and the other points in the commented PDF.**

*>> Thank you for the comments in the PDF. We respond to the more substantial comments here below in the online reply. The minor technical/editorial suggestions such as literature we missed are much appreciated and will all be considered and addressed during the revision phase. Thanks also for the suggestions to improve the visualization details, which we will also address in the revision.*

- **Sample of two catchments**

*>> The two catchments are indeed not particularly dry areas when one uses the definition of arid. However since drought is a relative term, droughts can occur and have impacts in wet regions as well. In this initial study, we focused on these two catchments because of the high data availability, in particular glacier data that we needed to constrain the model simulations. The two catchments are illustrative of the effects of glacier modelling strategy and threshold level method on future streamflow droughts in glacierised regions. We aim to apply the results of this study in further research to areas around the world that are more sensitive to anomalies in glacier melt.*

- **Glacier routine**

*>> Thanks for commenting on the understandability of the glacier routine together with the Seibert et al. (2017) paper. We hope that this can help settle the original concern by the Editor whether the description was sufficient.*

- **Snow routine and snow towers**

*>> Snow redistribution and snow towers are not accounted for in the model. The snow redistribution routine, which is included in the model version of Seibert et al. (2017) was not yet available in our model version. Since snow is not redistributed from the higher elevation zones, snow towers are present in the model simulations of both the historical and future period. Snow towers can influence the glacier retreat and also the snowmelt contribution to streamflow. In case snow is redistributed on the glacier, the glacier will melt slower than in our case where there is no additional supply of snow to the glacier. However, the timing of the glacier retreat is also influenced by the various other simplifications in the modelling of the glacier retreat. The storage of snow in snow towers could result in less snowmelt contribution at the end of summer. However, we checked the elevation zones where snow towers occur and compared the amount of SWE stored in the snow towers with the total discharge and found that the influence of snow towers on the streamflow simulation is small (negligible to a few percent). Moreover, for our drought analysis we used a threshold which is based*

*on the simulated streamflow and therefore the small effect of the snow towers is present in both the streamflow and the threshold and it won't affect our drought analysis. When snow redistribution would be taken into account, both streamflow and threshold would have slightly other values (possibly resulting in slightly other drought characteristics), but we expect the same main processes to take place. When comparing the historical and future period (with the HVT) a slight mismatch in streamflow regimes caused by the snow towers could occur, because snow towers are built in the historical period run, but in the future period simulation snow towers also melt in some elevation zones or have disappeared in the far future (2071-2100). However, we think that this effect of the snow towers on the drought analysis is negligible because of the small ratio between SWE and total discharge. We will discuss the snow towers and the possible implications in the discussion in the revised version.*

- **Calibration period**

*>> For both catchments we wanted to have at least a 10 years calibration period. For Wolverine this period was only available in the period 2005-2014. For Nigardsbreen a longer time series was available and we decided to use the first part of the time series for calibration and the last 10 years for validation. The two study regions are very far apart from each other and have different climate developments. We hence think the benefit of comparable periods is low compared to the benefit of optimal use of available data.*

- **Fig. 3 Wolverine**

*>> In Figure 3, the observed and simulated regime do indeed not agree very well. However, in this Figure 3 the regime based on observations is only calculated based on 3 years for Wolverine (due to data availability in the historical period), while the simulated regimes are calculated based on 30 years of data. The observed regime is therefore more sensible to extreme years or measurement errors. We can show the agreement between the regimes of observations and $Qsim_o$ for the calibration period (not $Qsim_{cm}$, since they are not available for the calibration period of Wolverine) (Figure below) as inset in Figure 3 in the revised version and explain Figure 3 better.*

[Figure]

- **Thresholds in Switzerland**

*>> Yes we are aware of studies using HVT to quantify drought in climate impacts studies, see e.g.:*

- *Lehner et al., 2006 - Estimating the impact of global change on flood and drought risks in Europe, A continental, integrated analysis*

- *Arnell, 1999, The effect of climate change on hydrological regimes in Europe: a continental perspective*
- *Prudhomme et al., 2014 - Hydrological droughts in the 21st century, hotspots and uncertainties from a global multimodel ensemble experiment*

*Thanks for the information on Switzerland using the last 10 years as a threshold/index baseline and thus essentially a moving threshold. We will extend the discussion on this.*

References:

Arnell, N. W. (1999). The effect of climate change on hydrological regimes in Europe: a continental perspective. *Global environmental change*, *9*(1), 5-23.

Lehner, B., Döll, P., Alcamo, J., Henrichs, T., & Kaspar, F. (2006). Estimating the impact of global change on flood and drought risks in Europe: a continental, integrated analysis. *Climatic Change*, *75*(3), 273-299.

Prudhomme, C., Giuntoli, I., Robinson, E. L., Clark, D. B., Arnell, N. W., Dankers, R., ... & Hagemann, S. (2014). Hydrological droughts in the 21st century, hotspots and uncertainties from a global multimodel ensemble experiment. *Proceedings of the National Academy of Sciences*, *111*(9), 3262-3267.

---

## Author Comment (AC3) · 4 Jul 2017

We would like to thank Lena Tallaksen for reading the paper and the feedback and remarks regarding the drought terminology.

The comments are in **bold**, our response in *italic.*

**The paper addresses an important topic related to the influence of glaciers on the flow regime in a future (warmer) climate, and drought in particular. My remarks relate primarily to the terminology used for defining drought and do not address the full paper as such.**

**Two different threshold approaches are employed; a threshold based on the historical period and a transient threshold approach, whereby the threshold adapts every year in the future to the changing regimes. In both cases, drought occurs when the discharge falls below the threshold. A daily variable threshold is used (80th percentile), defined based on a 30-day moving average time series. There is no seasonal distinction made and droughts can occur any time of the year as long as the flow is below the daily varying threshold.**

**The study, which is based on two catchments, projects "extreme increases in drought severity in the future" for the scenario HVT-D, i.e. a historical threshold combined with a dynamical glacier area. More specifically, the simulations show a lower peak flow and a shift towards an earlier melt peak, implying higher than normal flow early in the summer season and lower than normal flow towards the end of the melt period (ref. Figure 7). Accordingly, the projected increase in drought severity (from the time of the peak and onwards) is mainly caused by a change in the timing of the melt peak, or as stated in the paper, "by the regime shift due to a reduction in glacier area". (It is recommended to use the same scale on the y-axis for the different plots in Figure 7 to ease the comparison.)**

*>> We will use the same scale on the y-axis for Figure 7 in the revised version.*

**Both catchments have typically glacier flow regimes with low flows in winter and high flows in summer. Projected changes in flow seasonality in catchments with glaciers are strongly linked to changes in the snow regime with more precipitation falling as rain (rather than snow) and less snow accumulating (with the exception of some high altitude regions). Milder winters are projected to lead to earlier spring flood, a tendency that can already be observed for Norway (Wilson et al., 2010). Similar, warmer spring and summers are projected to lead to earlier and more glacier melt (as long as the glacier volume does not reduce too much). However, a shift in the timing or a reduction in the flow during the snow or glacier melt season is not associated with an increase in drought in these cold climate regions; neither by the snow/glacier research communities nor by water management. Rather, if focus is on drought, there is a concern that a longer snow free season combined with an increase in evapotranspiration may lead to increased drought in the following low flow period (e.g. Wilson et al., 2010). Glacierised catchments located in wet climates such as western part of Norway are further expected to be less prone to droughts in the future as compared to catchments located in drier climates.**

*>> Yes, it is true that research on this issue is more socially relevant in the more vulnerable drier regions of the world, where the dependence on glacier melt water components is higher. Norway and Alaska are used as case studies in this research to analyse the effects of methodological choices because of their good data availability. We hope to apply the outcomes of this research to more vulnerable regions and clarify the role of the reduction of the more reliable meltwater nowadays versus the more variable rainfall-runoff component in different climatic situations. We will clarify this in the revised manuscript.*

**The terms 'flood' and 'drought', as well as 'high flow' and 'low flow' periods are well defined concepts in hydrology, and I would strongly argue against using the term 'drought' for a period**

with relatively low flow during the high flow season or equivalent, 'flood' for a period with relatively high flow during the low flow season, merely based on their percentage deviations from the seasonal flow regime (and not their impacts). Rather, I suggest referring to these deviations as streamflow anomalies (or deficiencies for drought) as originally proposed by Stahl (2001) when introducing the daily varying threshold approach, and later elaborated in Hisdal et al. (2004). As highlighted in these studies, the variable threshold approach is adapted to detect streamflow deviations during both high and low flow seasons, and periods with relatively low flow during the high flow season are commonly not considered droughts. Still, lower than normal flows during high flow seasons may be important for later drought development.

*>> We understand that it might be confusing to use the term streamflow drought for anomalies in streamflow during the high flow season, although it does fit within the definition of below normal water availabilities relative to climatology. However, these high flow season streamflow droughts as they are defined with this method have been described as important as well, and have been studied in other studies that used a variable threshold level method (e.g. Van Loon et al., 2015, Fundel et al., 2013) or standardised indices like the Standardized Runoff Index and Standardized flow index (Shukla & Wood, 2008, Vidal et al., 2010). Especially global studies looking at future drought, such as Prudhomme et al. (2014), Van Huijgevoort et al. (2014) and Wanders et al. (2015), define drought compared to climatology everywhere around the world, regardless of definition issues. We also acknowledge that especially streamflow droughts with relative high deficits and long durations within the high flow season will affect downstream water users (e.g. Immerzeel et al., 2010, Messerli et al., 2004). So we do think that the term streamflow drought for severe deficits in the high flow season may have some merit, especially more downstream and for speaking to the global drought community. Moreover, we think that drought is not a so 'well defined concept in hydrology' when looking at the different uses of the term drought in literature and the numerous drought indices that exist. Within the group of authors we have discussed this definition issue extensively and although we have different opinions we settled for using the term streamflow drought in this paper for practical reasons. We suggest to make our definition of streamflow drought in this study more clear and explain how it differs from other studies in the introduction. We will also add some discussion about this drought definition issue in the revised version.*

References:

Fundel, F., Jörg-Hess, S., and Zappa, M. (2013). Monthly hydrometeorological ensemble prediction of streamflow droughts and corresponding drought indices, Hydrology and Earth System Sciences, 17, 395-407, doi:10.5194/hess-17-395-2013

Immerzeel, W. W., Van Beek, L. P., & Bierkens, M. F. (2010). Climate change will affect the Asian water towers. *Science*, *328*(5984), 1382-1385.

Messerli, B., Viviroli, D., & Weingartner, R. (2004). Mountains of the world: vulnerable water towers for the 21st century. *Ambio*, 29-34.

Prudhomme, C., Giuntoli, I., Robinson, E. L., Clark, D. B., Arnell, N. W., Dankers, R., ... & Hagemann, S. (2014). Hydrological droughts in the 21st century, hotspots and uncertainties from a global multimodel ensemble experiment. Proceedings of the National Academy of Sciences, 111(9), 3262-3267.

Shukla, S., & Wood, A. W. (2008). Use of a standardized runoff index for characterizing hydrologic drought. *Geophysical research letters*, *35*(2).

Van Huijgevoort, M. H. J., Van Lanen, H. A. J., Teuling, A. J., & Uijlenhoet, R. (2014). Identification of changes in hydrological drought characteristics from a multi-GCM driven ensemble constrained by observed discharge. *Journal of hydrology*, *512*, 421-434.

Van Loon, A. F., Ploum, S. W., Parajka, J., Fleig, A. K., Garnier, E., Laaha, G., & Van Lanen, H. A. J. (2015). Hydrological drought types in cold climates: quantitative analysis of causing factors and qualitative survey of impacts. Hydrology and Earth System Sciences, 19(4), 1993-2016.

Vidal, J.-P., Martin, E., Franchistéguy, L., Habets, F., Soubeyroux, J.-M., Blanchard, M., and Baillon, M.: Multilevel and multiscale drought reanalysis over France with the Safran-Isba-Modcou hydrometeorological suite, Hydrol. Earth Syst. Sci., 14, 459-478, doi:10.5194/hess-14-459-2010, 2010.

Wanders, N., Wada, Y., & Van Lanen, H. A. J. (2015). Global hydrological droughts in the 21st century under a changing hydrological regime. *Earth System Dynamics*, *6*(1), 1.

---

## Author Response (AR1)

Dear Editor,

We would like to thank the two reviewers, the editor and Lena Tallaksen for the feedback on our manuscript. Please find our detailed answers to the reviewers and the short comment and the changes we have made below. A track change version of the manuscript is also included. We clarified the manuscript where it was asked for by the reviewers and we restructured the discussion and adjusted some of the figures. Page and line numbers refer to the track change version of the manuscript.

Response to reviewer #1

The reviewer's comments are in **bold**, our response in *italic.*

**General comment**
**This paper is an analysis on the possible causes of streamflow droughts in glacierised catchments in the context of climate change (affecting glacier geometry, melt rate, discharge regime and drought). The authors have chosen 2 different highly (more than 60%) glacierised catchments for their study. They considered 2 different approaches for modeling glacier change: (1) glacier topography remains constant, and (2) glacier topography is empirically updated every year according to surface mass balance.**

**First, different approaches on how taking into account changes in glacier geometry in streamflow modeling is discussed. Since the approaches used are empirical, no ice dynamics in the strict sense is considered. So I suggest to modify ''glacier dynamics'' into ''glacier changes'' in the title.**

*>> The approaches to take into account changes in glacier geometry are indeed empirical and complex ice flow modelling is not used. We used the term dynamics to indicate that the glacier geometry change and streamflow modelling is coupled and therefore the glacier is adjusted in a dynamical way. However we agree that this may be confusing and we changed the "glacier dynamics" into "glacier changes" in the title.*

**Since glacier topography always changes in a changing climate, it is in my opinion not a sound option to analyze streamflow evolution without adapting the glacier topography accordingly. A more realistic option would be to assume no glacier at all. But assuming a constant and arbitrary glacier surface area in a changing climate will produce streamflow results which can hardly be interpreted.**

*>> We agree that glacier geometry (in the model described by glacier area and glacier thickness in each elevation zone) always changes in a changing climate and that one should account for that in the modelling of glacierised catchments to obtain realistic predictions, as we discuss in the manuscript. The benchmark simulation with static glaciers in our study can still serve a number of interesting purposes. Most importantly, we can use the static glacier modelling to isolate the direct effect of changes in temperature and precipitation from the effect changes in glacier extent, so that we can understand the corresponding changes in seasonal streamflow variability better. While the resulting streamflow quantities might not be realistic the benchmark can still be useful for increasing our understanding of how streamflow hydrograph changes will lead to changes in the processes leading to streamflow droughts. This benchmark method has also been used many times in hydrological modelling studies (see e.g. Akhtar et al., 2008; Stahl et al., 2008; Tecklenburg et al., 2012; Sun et al., 2015). Finally, some models still only use a constant glacier cover through time, either since they only model a short period of time or since the model does not allow to model glacier change (see e.g. Singh & Kumar, 1997; Klok et al., 2001; Shabalova et al., 2003; Verbunt et al., 2003; Singh & Bengtsson, 2005; Terink et al., 2005; Horton et al., 2006; Rahman et al., 2013).*

*We clarified these reasons for using a constant glacier area in section 3.1:*

*The glacier modelling options that are evaluated include a static and infinite glacier reservoir and a glacier area change conceptualisation using the Δh-parametrisation of Huss et al. (2010). These two glacier modelling options, in the following referred to as 'constant' and 'dynamic' glacier modelling options, are further explained in Sect. 3.2. Although the constant glacier modelling option will be unrealistic in transient mode we include this option in our analysis because dynamical glacier modelling is not yet included in all (large) scale hydrological models (e.g. Zhang et al., 2013) and it is an interesting benchmark, also frequently used in other studies (Akhtar et al., 2008, Stahl et al., 2008, Tecklenburg et al., 2012). The effect on streamflow drought characterisation has not yet been assessed.*

*Assuming no glacier is certainly a possible option; yet a less defendable one as the model parameters were calibrated to glacierised catchments and model parameters thus effective will reflect the typical sensitivities and relations among fluxes of glacierised catchments. Nevertheless, in our simulations for the Wolverine catchment a no glacier scenario also occurred during part of the simulation period and can happen in general in studies of future simulations of glacierised catchments. However, no solution to avoid this problem with calibrated model parameters and changing glaciers and long simulation periods does exist. We included the no glacier area option in the discussion:*

*Another option regarding the glacier modelling could be the full removal of the glacier. In theory, the comparison of simulated discharge without glaciers, with constant glaciers and with dynamical glaciers can give interesting information about the role of glaciers in causing or preventing streamflow droughts. For example, apart from distinguishing between the anomalies in glacier melt and glacier dynamics as causing factors of streamflow drought, also anomalies in snow melt and precipitation deficits in relation to streamflow droughts could be better assessed. However, model parameters are calibrated to discharges and glacier mass balances of glacierised catchments and therefore reflect the typical sensitivities and relations among fluxes for glacierised catchments. Hence, these parameters cannot be directly used to simulate a non-glacierised catchment. We therefore did not include this option explicitly in our study. Nevertheless, in our dynamical glacier conceptualisation we simulate a glacier disappearance for the Wolverine catchment from around 2060 onwards, while still using the same parameters. A solution, however, with time-varying parameters for simulation of long time periods and retreated glaciers does not yet exist (see e.g. Merz et al., 2011; Thirel et al., 2015; Heuvelmans et al., 2004; Paul et al., 2007; Farinotti et al., 2012).*

**Furthermore, I feel that the different ways of defining a drought a bit confusing. Since I am not hydrologist, I apologize for that. But in my opinion, it would be sufficient to define a reference period (RP) (f. i. 1960-1990 as used in Switzerland for climatology), and to present streamflow results outside this RP as deviations from it. I think results presented this way will be more useful for water management purposes.**

*>> We used and compared two methods to define streamflow droughts in our study: the Historical Variable Threshold (HVT) and Transient Variable Threshold (TVT). For the HVT we use a fixed RP period in the past, as the reviewer describes, and deviations from this Historical Threshold in the future (below threshold values) is what we define as streamflow droughts, in the HVT method. The aim of this paper was to compare this approach with another threshold approach, the TVT. This Transient Threshold changes according to the changing regime because there is no fixed reference period (it is a moving 30-year window). The results show that using a HVT or RP in changing catchments, like glacierised catchments, may not always be the most useful method for water management purposes since this method indicates changes in the regimes but not real changes in streamflow droughts. In our study, in the Nigardsbreen catchment for example, the HVT is based on a*

*period where the catchment was highly glacierised and using this threshold outside the reference period (in the future) where the glacier has retreated gives a large increase in drought deficit while in fact it is only a small shift in timing. The HVT method thus also leads to a strange comparison of different streamflow generation processes controlling the streamflow signal and variability (especially in the Wolverine catchment where the glacier has disappeared). We do agree that it is important to look at changes and deviations from a past RP, e.g. by looking at changes in the hydrological regime (see Fig. 5), but for water management we think it is also relevant to take these changes and adaptation into account and change the point of view and look at future streamflow variability to analyse streamflow droughts.*

*We added clarifications about the use of the two thresholds in Section 3.1 and Section 3.5 and introduce and further discuss the use of the two thresholds (e.g. for water management purposes) in the introduction and discussion (p.25 l. 9-11 and 18-23). We hope this will help the reader to understand the differences between the two threshold methods.*

**The paper is well written and the results well presented.**

**I can recommend publication.**

*>> Thank you for this positive evaluation.*

**Specific comments:**
**p. 18 line 14: "… in de left …'' needs correction**

*>> Thank you, this is corrected.*

Reviewer #2

The reviewer's comments are in **bold**, our response in *italic.*

**Dear authors,**

**I like the manuscript but I have a couple of concerns:**

**a) Glacier dynamics: C D: You are comparing a state-of-the-art approach (dynamical glaciers), with an approach which is acknowledged to be inadequate in transient mode (constant glaciers). I think you should have a look at the option without glacier change, but tax it as inadequate early enough in the paper (Figure 6) and continue with the dynamical model only. Up to Figure 6 you could also work with a version where you remove the glaciers from the beginning.**

*>> We agree that constant glaciers are not realistic if used for prediction and discuss this in the manuscript. As the reviewer notes, it is interesting to have a look at this option, since not all hydrological models (at all scales) do yet use a dynamical glacier approach and we want to show the effect of this glacier modelling choice on streamflow drought analysis. In the revised manuscript, we made clearer from the beginning that constant glaciers are not realistic in transient mode (in Section 3.1).*

*However, we do think this setting is useful and we therefore do not agree that we should only continue with dynamical glacier modelling from Figure 6 onwards. From Figure 6 onwards, especially in Figure 8, we show that modelling with constant glaciers is actually an interesting benchmark simulation. We can use it to isolate the effect of short term anomalies in precipitation and temperature from the effect of long term glacier changes on streamflow droughts and it thus gives more insight in the processes causing streamflow droughts in glacierised catchments (see also comment to reviewer #1). In relation to this, a version where we would remove the glaciers from the beginning would be a very interesting option, because it would give more insight to what extent streamflow droughts are caused by precipitation anomalies and/or snow melt anomalies (no glacier version), glacier melt anomalies (constant glacier version) and glacier dynamics (dynamical glacier option). However, as mentioned in the reply to reviewer #1, model parameters are calibrated to glacierised catchments and therefore reflect the typical sensitivities and relations among fluxes of glacierised catchments and cannot directly be used to simulate catchments without a glacier. We added this in the discussion (see indicated changes in the reply to reviewer #1).*

**b) HVT TVT: Here I fear that regime shift (shown by HVT) and actual drought-analysis (shown by TVT) are mixed up. In the discussion part this is presented as finding, but I think this should be stated from the beginning.**

*>> Interestingly, reviewer #1 had the opposite suggestion, of using only a HVT approach for future drought analysis. We mention in the introduction that using a HVT leads to severe 'droughts' in case of regime shifts (Van Huijgevoort et al., 2014). Despite this, HVT is used in studies to analyse future streamflow droughts at the global scale, which we also mention in the introduction. In the threshold level method droughts are defined as discharges below the threshold, whether HVT or TVT, so purely looking at the definition both are droughts. However, we do discuss this issue of regime shifts and whether regime-shift 'droughts' should also be interpreted as droughts. Given the confusion of both reviewers and the lack of consideration of this drought definition issue in most global future drought studies we infer that it is useful to discuss the effects of certain methods (HVT vs. TVT). We conclude (with the reviewer) that some are better suitable for the drought analysis than others, depending on the focus of a study/what the research interest is.*

*We included additional references that use a HVT to study climate change impacts on drought in the introduction (Page 3, line 29).*

**c) For calibration and validation I would use a seasonally varying discharge value instead of the average discharge as benchmark (see detailed comment in the commented manuscript)**

*>> This is a good point. We agree that Nash Sutcliffe criterion is not the best objective function to use in areas with strong seasonal discharge. It would indeed be better to use the monthly mean of the observations. However, such an objective function is not available (yet) within the calibration tool of the HBV-light model. We discuss this now in Section 5 (p.23 l.30-34). Also, the objective function that we used is not based on the whole discharge time series, but for 40% on glacier mass balances, 40% on seasonal discharge and 20% on the peak flows. We therefore do not calibrate on the whole seasonal cycle and avoid this problem to a certain extent. But still, using the monthly mean of the observations in the seasonal discharge calibration part would be beneficial.*

**Please clarify these issues and the other points in the commented PDF.**

*>> Thank you for the comments in the PDF. The more substantial comments are presented below. The minor technical/editorial suggestions such as literature we missed are much appreciated and have all been considered and addressed in the revised manuscript.*

-   **Sample of two catchments**

*>> The two catchments are indeed not particularly dry areas when one uses the definition of arid. However since drought is a relative term, droughts can occur and have impacts in wet regions as well. In this initial study, we focused on these two catchments because of the high data availability, in particular glacier data that we needed to constrain the model simulations. The two catchments are illustrative of the effects of glacier modelling strategy and threshold level method on future streamflow droughts in glacierised regions. We aim to apply the results of this study in further research to areas around the world that are more sensitive to anomalies in glacier melt. We included this in the discussion (p. 26 l. 32-34).*

- **Glacier routine**

>> *Thanks for commenting on the understandability of the glacier routine together with the Seibert et al. (2017) paper. We hope that this can help settle the original concern by the Editor whether the description was sufficient.*

- **Snow routine and snow towers**

>> *Snow redistribution and snow towers are not accounted for in the model. The snow redistribution routine, which is included in the model version of Seibert et al. (2017) was not yet available in our model version. Since snow is not redistributed from the higher elevation zones, snow towers are present in the model simulations of both the historical and future period. Snow towers can influence the glacier retreat and also the snowmelt contribution to streamflow. In case snow is redistributed on the glacier, the glacier will melt slower than in our case where there is no additional supply of snow to the glacier. However, the timing of the glacier retreat is also influenced by the various other simplifications in the modelling of the glacier retreat. The storage of snow in snow towers could result in less snowmelt contribution at the end of summer. However, we checked the elevation zones where snow towers occur and compared the amount of SWE stored in the snow towers with the total discharge and found that the influence of snow towers on the streamflow simulation is small (negligible to a few percent). Moreover, for our drought analysis we used a threshold which is based on the simulated streamflow and therefore the small effect of the snow towers is present in both the streamflow and the threshold and it won't affect our drought analysis. When snow redistribution would be taken into account, both streamflow and threshold would have slightly other values (possibly resulting in slightly other drought characteristics), but we expect the same main processes to take place. When comparing the historical and future period (with the HVT) a slight mismatch in streamflow regimes caused by the snow towers could occur, because snow towers are built in the historical period run, but in the future period simulation snow towers also melt in some elevation zones or have disappeared in the far future (2071-2100). However, we think that this effect of the snow towers on the drought analysis is negligible because of the small ratio between SWE and total discharge. We add a discussion on the snowtowers on p.24 l. 15-20.*

**Calibration period**

>> *For both catchments we wanted to have at least a 10 years calibration period. For Wolverine this period was only available in the period 2005-2014. For Nigardsbreen a longer time series was available and we decided to use the first part of the time series for calibration and the last 10 years for validation. The two study regions are very far apart from each other and have different climate developments. We hence think the benefit of comparable periods is low compared to the benefit of optimal use of available data.*

- **Fig. 3 Wolverine**

>> *In Figure 3, the observed and simulated regime do indeed not agree very well. However, in this Figure 3 the regime based on observations is only calculated based on 3 years for Wolverine (due to data availability in the historical period), while the simulated regimes are calculated based on 30 years of data. The observed regime is therefore more sensible to extreme years or measurement errors. We included the comparison of the hydrological regimes of observed, and simulated Q (forced by observations) for the calibration period as inset in Fig. 3c. Furthermore, we clarified the figure caption and the description in the Results Section.*

- **Thresholds in Switzerland**

>> *Yes we are aware of studies using HVT to quantify drought in climate impacts studies, see e.g.:*

- *Lehner et al., 2006 - Estimating the impact of global change on flood and drought risks in Europe, A continental, integrated analysis*
- *Arnell, 1999, The effect of climate change on hydrological regimes in Europe: a continental perspective*
- *Prudhomme et al., 2014 - Hydrological droughts in the 21st century, hotspots and uncertainties from a global multimodel ensemble experiment*

*And included the first two in the introduction, to clarify the use of HVT in our study (namely, HVT and TVT are both used in future drought studies, see also reply to comment c). Thanks for the information on Switzerland using the last 10 years as a threshold/index baseline and thus essentially a moving threshold. Unfortunately we could not find an application that indeed uses this 10-year reference for a transient look at time changes in low flows or drought, but only in the rule-set for the determination of Q347 where no long-term records are available (BAFU publications). Hence, it appears the motive is somewhat different and we did not yet include this in the discussion. Nevertheless, appropriate references to include this information in the revised manuscript would be welcome.*

**Short comments posted in PDF supplement:**

- Empirical quantile mapping reference

>> *We think it is beyond the scope of this paper to discuss bias correction, in this case for precipitation, of climate model data. We therefore did not include this reference.*

- Light blue areas in Fig 4.

>> *We changed and clarified the colors in the figure caption.*

- Running means and indications on inter annual variability, using Q50 and Q80, Fig5

>> *We show the Q80 already in Figure 6 and therefore did not include it in Figure 5. Regarding the use of Q50, we think that this does not give any additional information compared to the mean regime that we used in our manuscript. We applied the moving window smoothing suggestions in all panels, so that the changes we describe in the text are indeed better visible now in Figure 5. The comment on including an indication of the inter-annual variability of the historical period was not completely understood, but since we do not discuss it in the manuscript and the figure has become clearer due to the smoothing, we decided not to include inter-annual variability in the figure.*

- Threshold plot log scale

>> *In Fig. 6 we zoomed in to the winter low flow period and add this as inset to the plot, to be better able to distinguish the different thresholds.*

Response to short comment

We would like to thank Lena Tallaksen for reading the paper and the feedback and remarks regarding the drought terminology.

The comments are in **bold**, our response in *italic.*

**The paper addresses an important topic related to the influence of glaciers on the flow regime in a future (warmer) climate, and drought in particular. My remarks relate primarily to the terminology used for defining drought and do not address the full paper as such.**

**Two different threshold approaches are employed; a threshold based on the historical period and a transient threshold approach, whereby the threshold adapts every year in the future to the changing regimes. In both cases, drought occurs when the discharge falls below the threshold. A daily variable threshold is used (80th percentile), defined based on a 30-day moving average time series. There is no seasonal distinction made and droughts can occur any time of the year as long as the flow is below the daily varying threshold.**

**The study, which is based on two catchments, projects "extreme increases in drought severity in the future" for the scenario HVT-D, i.e. a historical threshold combined with a dynamical glacier area. More specifically, the simulations show a lower peak flow and a shift towards an earlier melt peak, implying higher than normal flow early in the summer season and lower than normal flow towards the end of the melt period (ref. Figure 7). Accordingly, the projected increase in drought severity (from the time of the peak and onwards) is mainly caused by a change in the timing of the melt peak, or as stated in the paper, "by the regime shift due to a reduction in glacier area". (It is recommended to use the same scale on the y-axis for the different plots in Figure 7 to ease the comparison.)**

*>> We now use the same scale on the y-axis for Figure 7 in the revised version.*

**Both catchments have typically glacier flow regimes with low flows in winter and high flows in summer. Projected changes in flow seasonality in catchments with glaciers are strongly linked to changes in the snow regime with more precipitation falling as rain (rather than snow) and less snow accumulating (with the exception of some high altitude regions). Milder winters are projected to lead to earlier spring flood, a tendency that can already be observed for Norway (Wilson et al., 2010). Similar, warmer spring and summers are projected to lead to earlier and more glacier melt (as long as the glacier volume does not reduce too much). However, a shift in the timing or a reduction in the flow during the snow or glacier melt season is not associated with an increase in drought in these cold climate regions; neither by the snow/glacier research communities nor by water management. Rather, if focus is on drought, there is a concern that a longer snow free season combined with an increase in evapotranspiration may lead to increased**

**drought in the following low flow period (e.g. Wilson et al., 2010). Glacierised catchments located in wet climates such as western part of Norway are further expected to be less prone to droughts in the future as compared to catchments located in drier climates.**

>> *Yes, it is true that research on this issue is more socially relevant in the more vulnerable drier regions of the world, where the dependence on glacier melt water components is higher. Norway and Alaska are used as case studies in this research to analyse the effects of methodological choices because of their good data availability. We hope to apply the outcomes of this research to more vulnerable regions and clarify the role of the reduction of the more reliable meltwater nowadays versus the more variable rainfall-runoff component in different climatic situations. We add this in the discussion (p. 26 l 32-34).*

**The terms 'flood' and 'drought', as well as 'high flow' and 'low flow' periods are well defined concepts in hydrology, and I would strongly argue against using the term 'drought' for a period with relatively low flow during the high flow season or equivalent, 'flood' for a period with relatively high flow during the low flow season, merely based on their percentage deviations from the seasonal flow regime (and not their impacts). Rather, I suggest referring to these deviations as streamflow anomalies (or deficiencies for drought) as originally proposed by Stahl (2001) when introducing the daily varying threshold approach, and later elaborated in Hisdal et al. (2004). As highlighted in these studies, the variable threshold approach is adapted to detect streamflow deviations during both high and low flow seasons, and periods with relatively low flow during the high flow season are commonly not considered droughts. Still, lower than normal flows during high flow seasons may be important for later drought development.**

>> *We understand that it might be confusing to use the term streamflow drought for anomalies in streamflow during the high flow season, although it does fit within the definition of below normal water availabilities relative to climatology. However, these high flow season streamflow droughts as they are defined with this method have been described as important as well, and have been studied in other studies that used a variable threshold level method (e.g. Van Loon et al., 2015, Fundel et al., 2013) or standardised indices like the Standardized Runoff Index and Standardized flow index (Shukla & Wood, 2008, Vidal et al., 2010). Especially global studies looking at future drought, such as Prudhomme et al. (2014), Van Huijgevoort et al. (2014) and Wanders et al. (2015), define drought compared to climatology everywhere around the world, regardless of definition issues. We also acknowledge that especially streamflow droughts with relative high deficits and long durations within the high flow season will affect downstream water users (e.g. Immerzeel et al., 2010, Messerli et al., 2004). So we do think that the term streamflow drought for severe deficits in the high flow season may have some merit, especially more downstream and for speaking to the global drought community. Moreover, we think that drought is not a so 'well defined concept in hydrology' when looking at the different uses of the term drought in literature and the numerous drought indices that exist. Within the group of authors we have discussed this definition issue extensively and although we have different opinions we settled for using the term streamflow drought in this paper for practical reasons.*

*We made our definition of streamflow drought clearer in the abstract and introduction. We included a short discussion on this drought definition issue and how our drought definition differs from other studies in the discussion (P. 26 l. 23-33).*

[revised manuscript text omitted]

---

## Author Response (AR2)

Dear Editor,

We thank the reviewers of the second round for the evaluation of our manuscript. Since reviewer #2 suggests to accept the paper as is, we will address here only the comments raised by reviewer #3. We have taken up the challenge to further improve the manuscript following the suggestions by reviewer #3, and although we don't agree with the reviewer on all points raised (see below), we do feel that the comments have helped us to further improve the readability and clarity. Our responses to the issues raised by referee #3 are indicated in *italics* below. A track change version of the manuscript is also included.

**Summary of the manuscript**
**This manuscript (ms) presents climate change projections of future streamflow droughts in two glacierized catchments. For this purpose the HBV model was used to project future runoff in two case studies (Wolverine in Alaska and Nigardsbreen in Norway) using two future climate change scenarios. Furthermore, four types of projections are presented in order to address the effects of glacier retreat and drought thresholds: i) simulations with constant glacier area, ii) simulations with dynamic glacier area (using the delta-h parametrization), iii) simulations with a drought thresholds (consecutive numbers of days with defined low discharge) defined in the reference period and iv) simulations with drought transient variable thresholds (TVT) defined for future periods. The study concludes that glacier dynamics and the threshold approach can significantly affect the assessment of future streamflow droughts.**

**Evaluation**
**In summary I think that an assessment of future stream flow droughts is extremely important for water managers and accordingly, I do think that the topic of the study is relevant. However, in my opinion there are some major concerns that should be addressed to make the study valuable:**

**- I am not sure why it is necessary to analyze the effects of constant glacier areas. To me this seems redundant, as it is evident that glacier areas will get smaller. I would rather like to see how much the glacier retreat affect streamflow drought. I recommend removing the simulations regarding constant glacier areas, and providing ice-melt contributions to runoff. In my opinion the simulations with constant glacier area could lead to misleading interpretations by some readers and diverts from the actually interesting topic: future stream flow droughts.**

*>> This point was already part of the first discussion round (see our second point in the reply to reviewer #1 and first point in the reply to reviewer #2). In short, we fully agree that modelling constant glaciers is not realistic, however, it is an interesting and useful benchmark (also used in many other studies), to compare the simulations with. We show that it can be used for analysing streamflow drought processes, to disentangle the effects of retreating glaciers and short term climate variability. Exactly this point is also mentioned by the reviewer: 'I would rather like to see how much the glacier retreat affect streamflow drought'. In order to answer this question a comparison is needed with a simulation where no glacier retreat takes place. By only looking at changing ice-melt contributions, changing weather patterns and/or decadal climate variability affecting discharge cannot be taken into account.*

*In the manuscript we now changed the way we presented the four scenarios (TVT-D, TVT-C, HVT-C and HVT-D), so that it is more clear for the reader that in order to study the effect of either changing baseline conditions or changing glacier area, we need to compare against a run in which this effect is not present. So one needs to either change the thresholds or change the glacier conceptualisation, to be able to analyse the effects of both (see matrix below).*

*We also added to the introduction that constant glaciers are used as benchmark in other studies.*

- **I am also not convinced why a TVT approach is helpful. For water managers a detailed assessment of future droughts based on present flow observations would be helpful. The TVT-approach might be misleading, as it may suggest that droughts will not be relevant in the future. I recommend focusing the results on streamflow drought based on the historic reference period, rather than investigating the effect of hypothetical numerical assumptions.**

*>> See the discussion in the first round (reviewer #1 point 3, reviewer #2 point 2). Similarly to the first issue raised (use of constant glacier areas) we argue that for a modelling experiment aimed at attribution and learning, besides using constant vs. changing glacier cover, it is also useful to test a constant versus changing threshold for the calculation of streamflow droughts.*

*We tested two existing threshold level methods in our study and compare and discuss them. We do not think that the TVT-approach is misleading, as it shows streamflow droughts from a future perspective, which are still relevant. The TVT aims to show the impact of drought under an adaptation scenario for both water managers and the ecosystem. There is a difference with the streamflow droughts indicated by the HVT-approach, but that doesn't mean one or the other is more appropriate. The droughts resulting from regime shifts extracted with the HVT-approach are more related to changed water availability and less to extreme below normal discharge events (see also the discussion in our manuscript). Overall, our aim is to show the effects of both thresholds in glacierised catchments, so that depending on the aim of a study one can choose the more appropriate threshold approach.*

- **The authors use discharge (Q) and glacier mass balances (MB) to calibrate their model; however they fail to present a validation of MB. For Q simulated and observed average daily values for the entire historic period are presented, making it impossible for the reader to judge if extreme events, i.e. droughts, are well reproduced by the model. Since an analysis of droughts is the objective of this ms, I believe a thorough discussion regarding the efficiency of extreme events should be discussed and visually presented.**

*>> We clarified the description of the observed and simulated mass balances in the calibration period in section 4.1. We also added two panels to Figure 3, to show the simulated MB in the calibration period (see Figure below).*

[Figure]

*We decided to present the average regime of the two catchments for a number of reasons. Firstly, climate model data should not be compared to observed data on a daily basis because the climate model only reproduces the statistics of the observed climate. Secondly, showing the entire time series would result in unreadable figures and the selection of a specific year would not be representative. Furthermore, the regime is commonly used for comparing simulations with observations and we also show that the inter-annual variability is well represented by the model. Finally, the KGE represents the correspondence of the daily values, so there is no need to show this in an additional figure.*

*We do, however, agree with the reviewer that an evaluation of how well droughts are reproduced by the model is important, because these are the objective of the study. We checked the drought characteristics in the observed streamflow and simulated streamflow for both catchments in the calibration period (see table below). Drought characteristics agree well for both catchments, only the number of droughts in the simulations of Nigardsbreen is a bit higher. In general we can see that the drought characteristics are close to the observed number, providing confidence in the (drought) simulations of the hydrological model.*

| | Droughts in | Number | Mean duration | Mean deficit | Intensity |
|---|---|---|---|---|---|
| Nigardsbreen (1967-2003) | Obs | 357 | 12.21 | 16.48 | 1.39 |
| | Sim - C | 565 | 9.66 | 12.27 | 1.23 |
| | Sim - D | 484 | 10.92 | 13.4 | 1.27 |
| Wolverine (2005-2014) | Obs | 99 | 13.49 | 25.97 | 2.80 |
| | Sim - C | 114 | 13.89 | 19.28 | 2.02 |
| | Sim - D | 99 | 12.95 | 25.73 | 2.64 |

**- Recent research has been focusing on estimating future rainfall, snow and ice runoff in glacierized catchments. In mountain areas snow melt has been identified as a dominant source of runoff generation, even in catchments with over 40% glacierization (how much glacierization is in the two case studies?). Accordingly, I would strongly recommend validating the snow cover (satellite data are available worldwide) before investigating streamflow droughts in mountain catchments. This is especially important for the HBV model, which accumulates snow height in higher altitudes to unrealistic heights if not calibrated adequately. The accumulation of snow height is of particular concern if simulations are run over several decades, as unrealistic snow heights can falsify the contribution of snow melt after few decades.**

*>> The amount of glacierization is described in the manuscript, see section 2.1, the second and third lines.*

*The issue of snow towers was also raised by reviewer #2 (see the reply). We checked our simulations and found that the effect of the snow towers in the discharge simulations is only minor. We therefore discuss that the effect of snow towers is considered to be small.*

**- Multi-objective function (pg2): the attribution of 40% to MB, 40% to Q in April-September and 20% to peak flow is arbitrary; what is the scientific rationale behind these weighing factors? I recommend giving all datasets equal weight. I also recommend providing the reader with the individual efficiencies for each part of the function (KGE might provide an overall efficiency, but is not suitable to discuss the efficiency regarding drought modelling, or glacier contribution, which are both essential to address climate change projections of stream flow droughts).**

*>> The choice of the weighing factors and which factors to include in the objective function is indeed always subjective. It depends on the focus of the study. For our streamflow drought definition the whole hydrograph is important. Discharge in winter is simulated well by the model due to low temperatures, so no special attention was needed for that during the calibration. We included in the calibration what was needed to get the best result. The main melt season is important, as well as the mass balance, as it helps to improve the modelling of the glacier component. We tested several objective functions and weighing factors and found that also taking into account the peak flows helps to improve the model simulations of the peak runoff caused by rain events.*

*The new figure and table in the manuscript now include the additional information about the model performance. Efficiency regarding drought modelling is shown in the drought characteristics table above (new Table 2 in revised ms) and glacier mass balance efficiency is shown as extra panels in (the previous) Figure 3 (now Figure 4 in the new ms).*

*The $R_{eff}$ values for the mass balances calibration are 0.51 and 0.83 for the dynamic glacier simulations of Nigardsbreen and Wolverine, respectively. The $R_{eff}$ of the seasonal calibration component ranges between 0.51 and 0.90 and the peak flow efficiency ranges from 0.15 to 0.60 for the two catchments and two glacier area conceptualisations. We added this in the manuscript.*

**- Climate scenarios (RCP4.5 and 8.5): since this study focuses on extreme events, i.e. droughts, it would be helpful to address how well the climate scenarios reproduce such extreme events. I recommend providing a table showing mean and standard deviation during QM correction.**

*>> Climate scenarios (RCP 4.5 and RCP 8.5) do not reproduce historical events, since they represent future projections. In addition, part of the climate model forcing data was obtained already bias corrected (for Nigardsbreen), so we cannot show pre and post bias correction information. Climate scenarios only include a control run for the past, which doesn't allow a comparison of individual events, but only the statistics. In the manuscript we do show that the discharge simulations with the climate data are close to the observations (observed discharge and discharge simulations forced by observed climate) (Figure 3 in old ms). The discharge is in our research the main variable of interest.*

**- Selection of case studies: why were two study sites chosen that are geographically so far apart? What is the scientific rationale behind the selection of the study sites?**

*>> The main reason for choosing these catchments is their relatively good data availability. We have made this clear now in the revised manuscript (study area section). Furthermore, we think it is interesting to show the effects of the approaches for different glacierised regions in the world, especially for catchments with a contrasting historical glacier behaviour (negative annual mass balances for Wolverine glacier and mainly positive annual mass balances for Nigardsbreen). The two catchments show a different response to climate change in the future (glacier disappearance or not)*

*and are therefore interesting case studies to show the effects of different threshold and glacier approaches.*

**- Conclusions: in my opinion the conclusions are flaw and do not really reveal new insights into drought dynamics. I recommend that the conclusions focus on future droughts based on present threshold (see comment above). By addressing all the concerns above, valuable conclusions could be generated, addressing the risk of future water shortages in the two case studies.**

*>> We kindly disagree with the reviewer and like to point out that the aim of this study is not to give future projections of streamflow droughts and addressing the risk of future water shortages in these two catchments, but rather showing the effect of different analysis approaches and systematically investigate how they can be used to analyse future streamflow droughts. From the discussion with all reviewers it also becomes clear that there is no consensus about the threshold method and how to use a glacier scenario and that this study could therefore be a useful addition to the scientific debate and progress with regard to this topic.*

**- Figures: please provide short titles (as done in fig 6) for all panels in all figures (next to the letters), this would help providing the reader an overview of the numerous sub-panels**

*>> We added the catchment names in the (old) figures 3 and 5 (new figures 4 and 6), as was done in Fig 6 (newly figure 7).*

**- Finally, I recommend to add a reflection why stream flow droughts are an important issue in the two selected case studies.**

*>> We mention in the discussion that streamflow droughts in terms of streamflow deficiencies can be important for downstream water users (e.g. ecosystems, water supply). All reviewers also acknowledge the importance of streamflow droughts in glacierised catchments. However, because of the definition we used to define droughts, not all our identified droughts will necessarily translate into impacts. Moreover, we would like to mention that this study is not an applied research about future projections in these two catchments, but rather a study developing a concept on how to study future streamflow droughts, which is also transferable to other glacierised catchments. We discussed in the previous round that we chose these catchments because of their good glacier data availability and that our aim for future research is to apply the outcomes of this work also to drier (and therefore more vulnerable?) glacierised regions in the world.*

**I leave it up to the editor and the readers of HESS to decide if the comments above should be implemented in the frame of revisions or in the frame of a new ms. All of the concerns above have been addressed in recent papers and I am convinced that this study could become a valuable contribution the water research if the concerns are addressed adequately.**

[revised manuscript text omitted]